# EFFICIENT ALLREDUCE WITH STRAGGLERS

## ABSTRACT

Distributed machine learning workloads use data and tensor parallelism for training and inference, both of which rely on the ALLREDUCE collective to synchronize gradients or activations. However, ALLREDUCE algorithms are delayed by the slowest GPU to reach the synchronization barrier before the collective (*i.e.,* the straggler). To address this challenge, we propose StragglAR: a parallel algorithm for ALLRE-DUCE that accelerates distributed training and inference by exploiting natural variation in GPU execution times. StragglAR implements a REDUCESCATTER among the remaining GPUs during the straggler-induced delay, and then executes a novel collective algorithm to complete the ALLREDUCE once the final GPU reaches the synchronization barrier. StragglAR achieves a $2\times$ theoretical speedup over popular bandwidth-efficient algorithms for large GPU clusters, surpassing the lower bound for bandwidth-optimal synchronous ALLREDUCE by leveraging the asymmetry in when GPUs reach the synchronization barrier. On an 8-GPU server, StragglAR provides a 25% speedup over state-of-the-art ALLREDUCE algorithms.

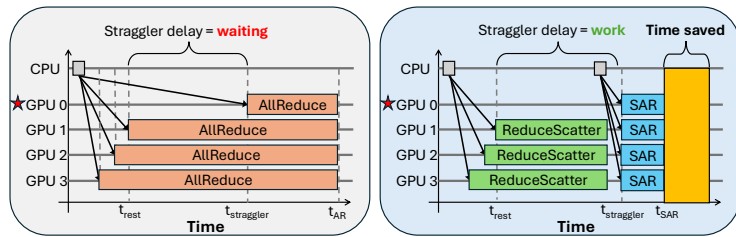

Figure 1: Straggler GPU (rank 0) causes all other GPUs to wait to begin the ALLREDUCE operation (left). Our proposed straggler-aware ALLREDUCE (right). Only communication kernels are shown.

## 1 INTRODUCTION

Distributed training and inference rely on collective communication primitives to exchange model gradients and activations across multiple GPUs. In particular, the ALLREDUCE primitive is used to average gradients across GPUs during data-parallel training and aggregate partial activations in tensor-parallel training and inference. ALLREDUCE and other communication primitives are implemented in collective communication libraries (CCLs), like NVIDIA's NCCL, which provide the core communication infrastructure for distributed ML. To enable efficient communication, CCLs implement ALLREDUCE using optimized parallel algorithms rooted in decades of research in high-performance computing (Dongarra et al., 2013). These algorithms are designed to minimize communication time across GPUs — a key bottleneck in scaling modern ML workloads with massive data transfer sizes. To meet this demand, CCLs use bandwidth-optimal algorithms (*e.g.,* Ring, Recursive Halving/Doubling), which heavily parallelize communication to fully leverage available inter-GPU network bandwidth.

**Our key insight.** Today's collective algorithms are built on a fundamental assumption: all GPUs initiate the collective *simultaneously*. However, in distributed ML, communication depends on preceding computation; the last GPU to finish the preceding computation, *i.e.,* the straggler, delays the entire ALLREDUCE (Fig. 1, left). Our experiments show that this happens regularly even within a *scale-up domain* — the high-performance multi-GPU servers (*e.g.,* NVIDIA DGX) and racks (*e.g.,* NVIDIA GB200) that are the workhorses of distributed ML. In our experiments fine-tuning Llama-3.2 (1B, 3B) within DGX servers, we observe straggler delays of up to 30 milliseconds

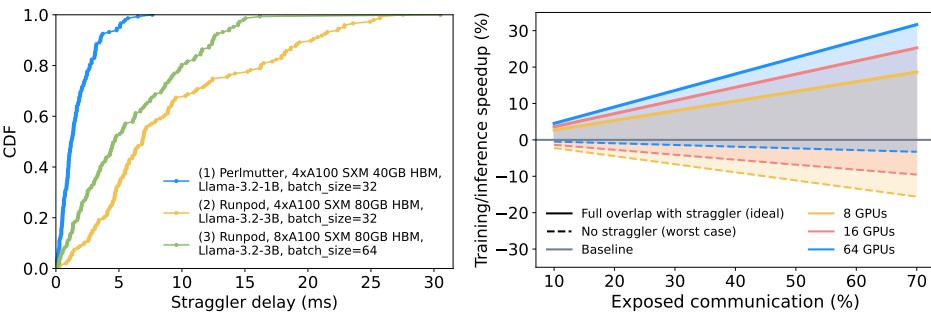

(a) Straggler delays in Llama-3.2 fine-tuning jobs.  (b) Range of simulated end-to-end speedups.

Figure 2: (a) CDFs of the straggler delay—the time between when the slowest and second-slowest ranks initiate the ALLREDUCE—in Llama-3.2 (1B, 3B) fine-tuning jobs on the Perlmutter supercomputer and RunPod VMs (with 3 independent runs per job). (b) End-to-end speedups for ML workloads using StragglAR compared to bandwidth-optimal algorithms (*e.g.,* Ring) for a 4 GB buffer (data-parallel buffer size of Llama-3.2-3B with local batch size of 4). Results are simulated with the $\alpha-\beta$ model using empirical $\alpha$ and $\beta$ values for the NVIDIA H100 DGX. Exposed communication is the percent of end-to-end time spent on ALLREDUCE. As cluster size scales, StragglAR shows large gains (+30%) while worst-case performance is on par with the baseline (-3%).

(Fig. 2a), prolonging the ALLREDUCE execution time and causing other GPUs to spend as much as 23-64% of their ALLREDUCE time idling. While stragglers in large datacenters are associated with network delays and hardware faults (Wu et al., 2024; Lin et al., 2025; Warraich et al., 2025; Jiang et al., 2024), we observe stragglers as an inherent challenge in distributed computation, where execution times naturally vary across GPUs. Existing mitigation strategies that approximate or drop the straggler's data can impact model convergence and do not generalize to ALLREDUCE in tensor-parallel training/inference. Thus, there remains a pressing need for bandwidth-efficient ALLREDUCE that is robust to stragglers while still preserving correctness.

**Our proposal.** Instead of treating stragglers as an anomaly, we propose designing algorithms that expect and exploit them. We design a novel ALLREDUCE algorithm, StragglAR (Fig. 1, right), that provably transmits up to $2\times$ fewer bytes than the known bandwidth-optimal lower bound by exploiting natural variation in GPU execution times. StragglAR leverages the straggler delay to perform useful communication, eagerly executing a REDUCESCATTER among the other GPUs. However, the resulting asymmetry between the data buffers of the straggler and non-straggler GPUs after the REDUCESCATTER introduces new challenges in implementing highly parallel communication that can outperform known bandwidth-optimal algorithms. Thus, we design a new parallel communication algorithm that utilizes this asymmetry to perform faster ALLREDUCE, achieving provably lower communication complexity in straggler settings while closely matching the complexity of bandwidth-optimal algorithms *even without stragglers* (Fig. 2b). **To our knowledge, our work is the first to show that the decades-old lower bound for bandwidth-optimal ALLREDUCE (Patarasuk and Yuan, 2009; De Sensi et al., 2024) can be surpassed by leveraging variation in compute times.**

Our hardware experiments on 8-GPU servers show that StragglAR achieves a 25% speedup over bandwidth-optimal ALLREDUCE algorithms (*e.g.,* Ring) in the presence of stragglers, and delivers end-to-end training speedups across multiple ML models. Simulations at larger scales show that these improvements grow with cluster size, reaching up to $2\times$ speedups at scale. As shown in Fig. 2b, StragglAR's performance range shifts upwards with cluster size, yielding clear upside with variation in execution times and minimal downside otherwise. Finally, this work opens a new design dimension for collective algorithms: temporal asymmetry. For decades, we have pursued spatial optimizations like topology-aware routing (Shah et al., 2023) and spectral optimizations (*e.g.,* compression), but we have insisted on temporal symmetry (*i.e.,* all GPUs start the collective together). Breaking this assumption not only provides an algorithmic approach to mitigating straggler effects, but also presents an opportunity to fundamentally redesign the collective algorithms that underpin distributed ML.

## 2 BACKGROUND AND RELATED WORK

In distributed ML, GPUs aggregate gradients and activations using the ALLREDUCE collective communication primitive (Figure 3), which transmits and reduces data across the inter-GPU network.

Data parallelism uses ALLREDUCE to average gradients, while tensor parallelism invokes it many times per model pass to exchange activations (Shoeybi et al., 2019a). Collective communication libraries (CCLs) like NCCL implement these primitives, choosing bandwidth-optimal algorithms (*e.g.,* Ring) for the large buffer sizes common in modern ML workloads (Shah et al., 2023). CCLs adopt a bulk-synchronous model where all GPUs must synchronize before the collective, but performance degrades when a straggler—the slowest GPU—delays synchronization (Warraich et al., 2025; Dean and Barroso, 2013; Wang et al., 2024a; Gangidi et al., 2024; Li et al., 2014). While stragglers result from inherent heterogeneity in GPU execution times, *severe* stragglers can stem from hardware issues (thermal throttling, power supply) or runtime factors (network congestion, compute skew) (Wu et al., 2024; Jiang et al., 2024; Grattafiori et al., 2024; Xiong et al., 2024). Recent work has highlighted the acute impact of stragglers at datacenter scale (Jiang et al., 2024; Lin et al., 2025), and our experiments show intrinsic 30 ms delays even within multi-GPU servers (Fig. 2a).

**Related Work.** Prior straggler mitigation strategies either identify and remove stragglers (Jiang et al., 2024; Lin et al., 2025), wasting compute and only addressing severe cases, or approximate/drop the straggler's data (Warraich et al., 2025; Harlap et al., 2016; Karakus et al., 2017; Recht et al., 2011), limiting applicability to data-parallel training and affecting convergence. Systems approaches (Wu et al., 2024; Zhao et al., 2024a) adapt workload placement or select among known algorithms, *e.g.,* AdapCC (Zhao et al., 2024a) uses known Tree algorithms that sacrifice bandwidth for latency. In contrast, StragglAR is a novel bandwidth-efficient ALLREDUCE algorithm that preserves exact reductions for both data and tensor parallelism and treats stragglers as intrinsic to distributed computation. While recent works synthesize new col-

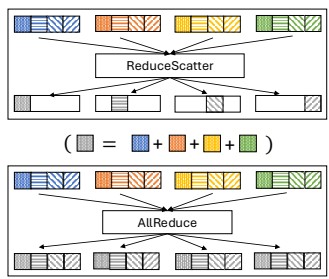

Figure 3: Collective operations.

lective algorithms for heterogeneous networks (Shah et al., 2023; Zhao et al., 2024a; Wang et al., 2020; Won et al., 2024; Zhang et al., 2024), in homogeneous scale-up domains these converge to classical bandwidth-optimal algorithms like Ring (Shah et al., 2023; Wang et al., 2020), which NCCL implements (Hu et al., 2025). We provide an in-depth discussion of related work in § A.

## 3 STRAGGLAR: A STRAGGLER-AWARE ALLREDUCE ALGORITHM

We now present StragglAR, a novel algorithm to speed up ALLREDUCE in the presence of stragglers.

$\alpha$-$\beta$ **cost model.** ALLREDUCE algorithms are typically analyzed using the $\alpha-\beta$ model of collective communication (Hockney, 1994; Thakur et al., 2005; Shah et al., 2023; Won et al., 2023; Wang et al., 2025). As per this model, sending a message of $s$ bytes takes $\alpha + s\beta$ time, where $\alpha$ is the fixed startup cost per message (independent of message size) and $\beta = \frac{1}{bandwidth}$ is the per-byte transmission cost.

Collective algorithms perform a series of data transfers among GPUs to achieve the desired operation. These algorithms specify a *schedule* that dictates which GPUs exchange data in each discrete time step, called a *round*. The $\alpha$ cost of an algorithm corresponds to the number of rounds, and the $\beta$ cost corresponds to the total number of serialized bytes sent (number of rounds × bytes sent per round). We use the term *buffer* to denote the communication volume in the ALLREDUCE call.

The goal of a collective algorithm is to provide a schedule that minimizes the total $\alpha-\beta$ cost. CCLs use $\alpha$-optimal algorithms for small buffers, when latency dominates, and $\beta$-optimal algorithms for larger buffers, when bandwidth becomes the bottleneck (as only $\beta$ cost scales with buffer size). Different algorithms affect the *coefficients* on $\alpha$ and $\beta$ while the $\alpha$ and $\beta$ constants come from the hardware. Bandwidth-optimal algorithms (*e.g.,* Ring) bound $\beta$ cost at $\sim 2s\beta$ regardless of the number of GPUs. StragglAR, like standard ALLREDUCE algorithms used in distributed ML, focuses on optimizing $\beta$ cost because today's large models incur large buffers of many MB to several GB.

**Key insights.** In standard ALLREDUCE algorithms, all GPUs must wait for the final GPU, the straggler, before starting the collective (GPU 0 in Fig. 1). In contrast, StragglAR uses this delay to perform a REDUCESCATTER (Fig. 3) among non-straggler GPUs. By productively using the straggler's delay, StragglAR reduces the data transfer cost once the straggler is ready.

Once the REDUCESCATTER is complete, StragglAR initiates a schedule of data transfers among all GPUs (straggler and non-stragglers) to complete the ALLREDUCE. However, even with the

---

**Algorithm 1:** StragglAR Schedule Generator for ALLREDUCE (Power-of-2 World Size)

---

**Input** : $n = 2^k$ GPUs, with rank $\sigma = n-1$ as the straggler (without loss of generality)
**Output** : schedule of chunk exchanges completing ALLREDUCE in $n-2 + \log n$ rounds

1   **Initialization:** rank $g < n-1$ holds partially reduced chunk $c_g$; straggler $\sigma$ holds none.
2   $A$: {}, dictionary of active chunks to the set of GPUs that hold it
3   schedule: [], list of sets of matchings over rounds to return
4   **for** *round* $r = 0$ **to** $n-2 + \log n$ **do**
5      **if** $r < n-1$ **then**
6         **Match** rank $r \leftrightarrow \sigma$: exchange chunk $c_r$; mark $r$ and $\sigma$ as unavailable
7      **if** $0 < r < \log n$ **then**
8         **Match** rank $r-1 \rightarrow$ rank $r-1+ \log n$: send chunk $c_{r-1}$
9         Each rank with a reduced chunk sends to any rank $g > 2(\log n - 1)$ without a chunk
10      **else**
11         $P_r \leftarrow$ available ranks holding oldest active chunk $\min_c A$
12         $Q_r \leftarrow$ available ranks holding other active chunks in $A \setminus \{\min_c A\}$
         `// Handle ranks in the critical window`
13         **for** $g = r+1$ *to* $r+\log n$ **do**
14            Select $c = \min\{c' \mid g$ lacks $c', \forall h \in A[c'], h \notin [r+1, r+\log n-1]\}$
15            **Match** rank $g \leftrightarrow h$; $g$ sends $c$; $h$ sends $c'$; mark $g$ and $h$ as unavailable
16      **Match** $p_i \leftrightarrow q_i$ for $p_i \in P_r, q_i \in Q_r$ to exchange their active chunks
17      Append a list of round $r$'s matchings to schedule
18      **if** $r \geq \log n$ **then**
19         $A[c_{r-\log n}] \leftarrow \emptyset$          `// c_{r-log n} is now fully propagated`
20      **if** $r < n-1$ **then**
21         $A[c_r] \leftarrow \{r\}$             `// Add newly active chunk to A`
22      For other active chunks $c$ exchanged: $A[c] \leftarrow A[c] \cup \{$receivers of $c$ in this round$\}$

---

precondition, designing an algorithm faster than $\beta$-optimal baselines is challenging because the precondition introduces inherent asymmetry (straggler vs. other ranks) that is difficult to parallelize. Thus, StragglAR designs communication to maximize parallelism in every round, achieving faster ALLREDUCE schedules than baselines. While computing efficient schedules is known to be combinatorially hard (Shah et al., 2023), StragglAR uses symmetry-breaking to quickly find them in polynomial time. For the desired multi-GPU setup, StragglAR is run once offline to generate ALLREDUCE schedules, which are then implemented to run in real-time without modification.

StragglAR is agnostic to the cause of the straggler or detection method (§4). Instead, it is a *fundamental parallel algorithm that communicates* $2\times$ *fewer bytes* than the known lower bound for ALLREDUCE during exposed communication in settings where overlap is possible, while retaining strong worst-case performance. We now describe the runtime environment StragglAR relies on.

**GPU cluster topology.** Like classical ALLREDUCE algorithms (*e.g.,* Ring, Recursive Halving/Doubling (Thakur et al., 2005)), StragglAR targets the **scale-up domain**—nowadays extending to 10-100s of GPUs (NVIDIA, 2024)—due to its homogeneous, any-to-any topology.

**GPU-GPU connectivity.** We assume each GPU has a single connection to the inter-GPU network, as is typical in modern switched scale-up domains (NVLink & NVSwitch). Thus, a GPU can fully utilize bandwidth by sending data to one peer at a time (confirmed empirically in §4). While multidimensional topologies (Jouppi et al., 2023) enable using multiple links in parallel, this proportionally reduces the available per-link bandwidth in typical NVSwitched configurations (as we show in §4).

### 3.1 ALGORITHM DESIGN

We consider a setting with $n$ ranks, $0, \ldots, n-1$. Each rank has a buffer of $s$ bytes that is evenly divided into $n-1$ chunks, each of size $\frac{s}{n-1}$ and indexed as $c_0, \ldots, c_{n-2}$. Without loss of generality, we assume rank $n-1$ is the straggler in this section; however, by symmetry, the algorithm applies regardless of which rank is the straggler (as further shown by our evaluation in §4).

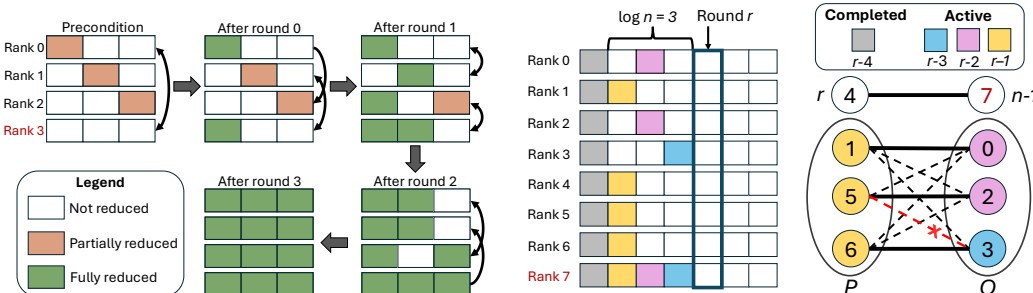

(a) Example StragglAR schedule for 4 GPUs.    (b) Example matching process, round $r=4$ with 8 GPUs.

Figure 4: Algorithm design (straggler in red). In (b), ranks 3 and 5 cannot match: 5 is in the critical window and 3's chunk was just unlocked. Pairing them would prevent 3's chunk from doubling.

**Precondition and postcondition.** StragglAR uses a precondition where the $n-1$ non-straggler ranks have completed a REDUCESCATTER among themselves, ideally overlapped with the delay of the straggler. This is feasible, since REDUCESCATTER is inherently $2\times$ faster than ALLREDUCE and is efficiently implemented by CCLs (Thakur et al., 2005). We refer to the reduced chunks after this step as **partially reduced** because they contain all but the straggler's data. StragglAR computes an optimized communication schedule to transition from this precondition to the standard ALLREDUCE postcondition, where all ranks, including the straggler, possess a fully reduced buffer.

The communication schedule consists of multiple rounds, each represented by a set of matchings, similar to some broadcast algorithms (Bar-Noy et al., 2000). In each round, matchings specify the pairs of ranks that communicate and which data chunks they exchange simultaneously. Each rank participates in exactly one matching per round. Since we fix the chunk size per round, the only lever to minimize communication time is to reduce the number of rounds. At the very least, $n-1$ rounds are required to transmit all $n-1$ chunks. StragglAR completes the ALLREDUCE in $n+\log n-2$ rounds, yielding a $\beta$ cost of $\frac{n+\log n-2}{n-1}s\beta$, which is less than the known lower bound of $2\frac{n-1}{n}s\beta$ (see §3.2). Alg. 1 summarizes the schedule computation when $n$ is a power-of-two, with modifications for non-power-of-two cluster sizes in §E. Fig. 4a visualizes the StragglAR schedule for $n=4$.

**Straggler pairings.** In round $r$, rank $r$ pairs with the straggler to fully reduce $c_r$. After $n-1$ rounds, the straggler's buffer is fully reduced and each chunk $c_i$ has been fully reduced on at least one non-straggler rank. Hence, $c_r$ can propagate only from round $r+1$ onward, once fully reduced.

**Phase 1.** In the first $\log n$ rounds, every rank obtains a fully reduced chunk, guaranteed by two rules:

- In round $r \in [1, \log n-1]$, rank $r-1$ sends chunk $c_{r-1}$ to rank $r-1+\log n$.
- Any other rank with a fully reduced chunk can send it to any other rank $g > 2(\log n-1)$.

**Phase 2.** At the start of round $r = \log n$, every rank has exactly one fully reduced chunk. Exactly one rank, $r-1$, has $c_{r-1}$. Two ranks have $c_{r-2}$ because it was first fully reduced in round $r-2$ and then transmitted to another rank in round $r-1$. By the same pattern, four ranks have $c_{r-3}$, *etc.*

**Definition.** An **active chunk** $c_j$ is a chunk that has been fully reduced with the straggler by round $r > j$, but has yet to propagate to all $n$ ranks.

As we prove in §D, every non-straggler possesses *exactly one* active chunk at any point in time. StragglAR aims to propagate active chunks to all ranks as quickly as possible. Each active chunk requires a minimum of $\log n$ rounds to propagate fully, by doubling in each round, so an active chunk $c_j$ is **due** for full propagation to all ranks by round $j + \log n$.

Algorithm 1 enables every active chunk to double in every round by ensuring that every rank transmits its active chunk and receives a different active chunk, with the oldest active chunk expiring in each round. We model each round as a bipartite matching problem between two disjoint sets of non-straggler ranks: $P_r$, ranks whose active chunk in round $r$ is $c_{r-\log n}$ and $Q_r$, the others. By the doubling property, it follows that $|P_r| = 2^{\log n-1} = \frac{n}{2}$ and thus $|Q_r| = n-1-|P_r| = \frac{n}{2}-1$.

This allows us to define an invariant: *For every round $r \in [\log n, n-2]$, rank $r \in P_r$.* This guarantees the $\log n$ propagation deadline, by ensuring that ranks receiving a new active chunk (by pairing with

| Algorithm | Best case | | Worst case | |
|---|---|---|---|---|
| | Latency | Bandwidth | Latency | Bandwidth |
| Ring | $2(n-1)\alpha$ | $\frac{2(n-1)}{n}s\beta \approx 2s\beta$ | $2(n-1)\alpha$ | $\frac{2(n-1)}{n}s\beta \approx 2s\beta$ |
| RHD | $2(\log n)\alpha$ | $\frac{2(n-1)}{n}s\beta \approx 2s\beta$ | $2(\log n)\alpha$ | $\frac{2(n-1)}{n}s\beta \approx 2s\beta$ |
| StragglAR$^\dagger$ | $(n+\log n-2)\alpha$ | $\frac{n+\log n-2}{n-1}s\beta \approx s\beta$ | $(2(n-2)+\log n)\alpha$ | $\frac{2(n-2)+\log n}{n-1}s\beta \approx 2s\beta$ |

Table 1: ALLREDUCE communication complexity. Because Ring and RH/D are agnostic to stragglers, their best-case and worst-case performance is identical. For our method, StragglAR$^\dagger$, the best-case bound is achieved when the straggler delay exceeds the initial REDUCESCATTER execution time while the worst-case bound is achieved when there is no straggler delay at all.

the straggler) have already received the chunk due in that round ($c_{r-\log n}$). Thus, we exclude rank $r$ from the matching process since it pairs with the straggler, enabling $|P_r| = |Q_r| = \frac{n}{2}-1$.

Now, we have two equal-sized, disjoint sets of ranks, $P_r \setminus \{r\}$ and $Q_r$. However, arbitrary matchings between the two sets risk violating the invariant *in the future*. In particular, ranks in the next $\log n$ rounds (the **critical window**) cannot receive chunks that would remain active when they must pair with the straggler (Fig. 4b). For example, rank $r+1$ may only receive $c_j$ with $j \leq r+1-\log n$. To enforce this, we first match ranks $j \in [r+1, r+\log n]$ in the critical window with partners outside this window that have the oldest available active chunk. After finalizing these matchings, the remaining ranks can be paired arbitrarily since every $u \in P_r$ and $v \in Q_r$ lack each other's active chunks.

After $n-1$ rounds, every chunk has been fully reduced, and the straggler has fully reduced its buffer. To fully propagate remaining chunks quickly, the straggler can be paired with arbitrary ranks, but always sends the final chunk. (This enables the straggler to be added to $Q_r$ so that $|P_r| = |Q_r| = \frac{n}{2}$.)

## 3.2 COMMUNICATION COMPLEXITY

**Theorem 1.** ALLREDUCE *schedules generated by StragglAR complete in* $n+\log n-2$ *rounds.*

We provide the proof in §D. Intuitively, each chunk $c_r$ propagates within $\log n$ rounds of being fully reduced. The final chunk $c_{n-1}$ only requires $\log n-1$ rounds to propagate, as the straggler can also help transmit it, thereby achieving $n+\log n-2$ total rounds. Achieving this bound requires matching ranks optimally for the current round without compromising on future propagation, where special care must be taken due to the pre-determined straggler pairings.

**Ideal performance.** When the straggler is delayed long enough to mask the REDUCESCATTER precondition (supported by Fig. 2a), StragglAR's exposed communication time consists of $n+\log n-2$ rounds with $\frac{s}{n-1}$ bytes sent per round. Thus, with sufficient straggler delay, the total time taken is

$$T_{\text{SAR}}^+ = (n + \log n - 2)\,\alpha + \frac{n + \log n - 2}{n - 1}\,s\beta.$$

Under these conditions, StragglAR achieves much lower $\beta$ cost than today's known bandwidth-optimal lower bound (see Table 1), and this advantage only grows as $n$ (number of GPUs) scales. We design StragglAR specifically for settings with stragglers, so this bound represents the performance under ideal use cases of our algorithm. The scaling behavior of known $\beta$-optimal algorithms is $\lim_{n\to\infty} \frac{2(n-1)}{n}s\beta = 2s\beta$, whereas StragglAR achieves $\lim_{n\to\infty} \frac{n+\log n-2}{n-1}s\beta = s\beta$, implying $2\times$ speedups in large-scale settings with a straggler. While $\alpha$ cost is typically negligible for large buffers, StragglAR scales better in $\alpha$ cost than Ring but more poorly than RHD.

**Worst case.** StragglAR exhibits its lowest performance when *none* of the REDUCESCATTER precondition is overlapped with the straggler delay (*i.e.,* no straggler delay, multiple simultaneous stragglers, *etc.*). A standard Ring REDUCESCATTER with $n-1$ ranks takes time $T_{RS} = (n-2)\alpha + \frac{n-2}{n-1}s\beta$. With no straggler delay, StragglAR serially executes the REDUCESCATTER followed by the custom schedule during the exposed communication time, resulting in worst-case complexity:

$$T_{\text{SAR}}^- = T_{RS} + T_{\text{SAR}}^+ = \left(2(n-2) + \log n\right)\alpha + \frac{2(n-2) + \log n}{n-1}\,s\beta.$$

Even in the worst case, where none of the REDUCESCATTER can be overlapped (*e.g.,* no stragglers), StragglAR remains competitive. StragglAR's worst-case scaling behavior is $\lim_{n\to\infty} \frac{n-2}{n-1} s\beta + \frac{n+\log n-2}{n-1} s\beta = 2s\beta$. Thus, at large scales, StragglAR's worst-case performance (*i.e.,* no straggler delay) mirrors that of baselines, which also have $2s\beta$ bandwidth complexity (see Table 1).

StragglAR's performance will fall within the range between the ideal and worst-case bounds, depending on the straggler delay. We note that the exact worst-case scenario is highly unlikely because GPU execution times are continuous: the likelihood of two GPUs completing exactly simultaneously would be near-zero, as corroborated by Fig. 2a. In summary, StragglAR offers substantial $2\times$ speedups over bandwidth-optimal baselines with sufficient straggler delay, and performs on par with baselines at scale even in the worst case with no straggler delay. We provide empirical and quantitative analyses of the *critical delay* required for StragglAR to provide speedups in §4 and §B, respectively. Our analysis in §B reveals that the critical delay decreases as $n$ increases and is approximately 0 for large $n$, hence why StragglAR offers competitive performance at scale even with no straggler delay. For smaller deployments, the critical delay will be non-zero and depends on the hardware characteristics, as we show in §4 and §B.

## 4 EXPERIMENTS

We implement the StragglAR algorithm to compute schedules, as well as a CUDA runtime that executes these schedules in multi-GPU setups.

**Detecting stragglers.** One advantage of StragglAR is that its initial REDUCESCATTER can be eagerly executed as soon as the first $n-1$ ranks are ready. If there is sufficient delay between the slowest rank and these $n-1$ ranks, StragglAR achieves its ideal performance; otherwise, performance falls in the range between the ideal and worst-case bounds discussed in §3 and §4.3. To consistently support ideal-case performance, the runtime can leverage online straggler detection tools (Zhao et al., 2024a) and pass the straggler's rank to the backend for conditional execution of the appropriate schedule. In our end-to-end experiments (§4.2), we fix the rank that StragglAR assumes to be the straggler (by profiling the workload ahead of time and picking a likely straggler rank, see §4.2). This stress-tests StragglAR, as there are many iterations in which the algorithm encounters its worst-case performance (*i.e.,* no precondition overlap), when a different rank is the straggler or there is no straggler at all.

**Schedule generator.** We implement StragglAR in Python to generate ALLREDUCE schedules. The algorithm takes the number of GPUs as input and outputs a schedule with matchings, specifying for each round which GPUs communicate and which chunks to send/receive. Schedule generation is run once offline and is fast—StragglAR computes the schedule for a 256-GPU cluster in <1.04 seconds.

**Runtime.** We implement the StragglAR schedules using the NCCL Point-to-Point (P2P) API (NCCL) (also used in prior work (Wang et al., 2020)), for GPU communication, combined with custom kernels for reduction. After completing `ncclReduceScatter()` among non-straggler GPUs, we invoke the runtime to execute the StragglAR schedule. We package both REDUCESCATTER and StragglAR into an API that has the same functionality as `ncclAllReduce()`.

**Baselines.** Given the homogeneous, switched connectivity of modern scale-up networks, we compare StragglAR to the strongest bandwidth-optimal ALLREDUCE algorithms for this setting. Our baselines are **(1) Ring** (Patarasuk and Yuan, 2009; NVIDIA, 2024a) and **(2) Recursive Halving/Doubling (RHD)** (Bruck et al., 1994), which are the gold-standard bandwidth-optimal algorithms in homogeneous networks (Hu et al., 2025); **(3) MSCCL** (Cowan et al., 2023), the only recent work we are aware of to synthesize a new algorithm for scale-up domains; and **(4) Broadcast**, a naive straggler-aware baseline. The MSCCL baseline uses their *allpairs* algorithm: a one-round REDUCESCATTER where each GPU splits its bandwidth across all peers, followed by a one-round mirror ALLGATHER. This achieves the same $2s\beta$ bandwidth cost as classical algorithms but reduces latency to $2\alpha$ on switched networks. The Broadcast baseline is straggler-aware, as non-stragglers complete an ALLREDUCE during the straggler delay and then finish with a pairwise-exchange broadcast once the straggler arrives. More details on baselines can be found in §F. For fair comparison of the algorithmic contribution, we implement baselines using the NCCL P2P API and the same CUDA compute kernels as StragglAR. While recent works such as MSCCL++ (Shah et al., 2025) provide alternatives to NCCL for finer-grained development, this approach is complementary to ours, as MSCCL++ provides a novel API to *implement* collective algorithms rather than designing new algorithms themselves.

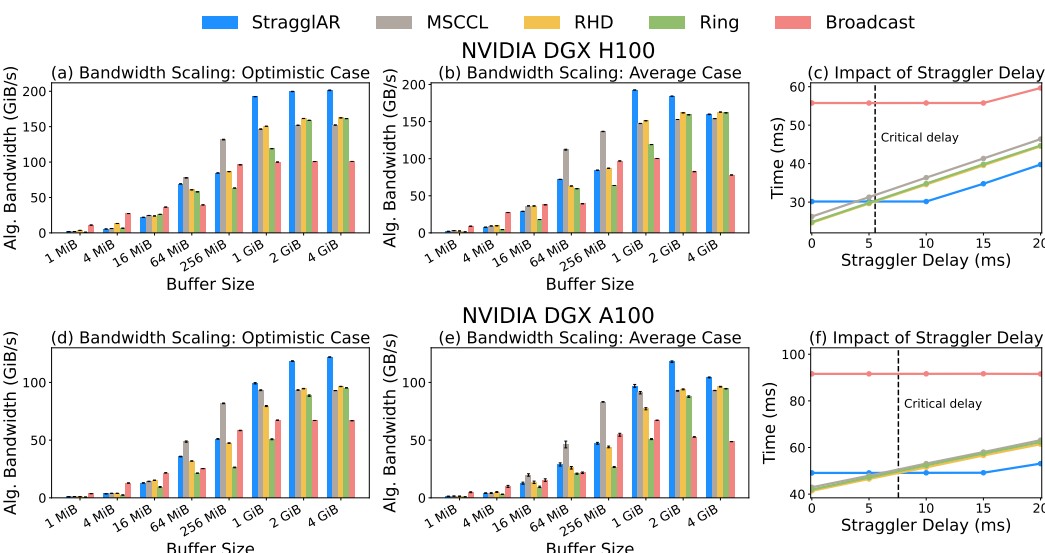

Figure 5: Benchmarking ALLREDUCE performance for different algorithms on NVIDIA DGX H100 (a-c) and A100 (d-f) servers. (a), (d) show the optimistic use case, by assuming REDUCESCATTER can complete within the straggler delay. (b), (e) use the environment-specific average straggler delay from the Llama-3.2-3B training experiments of 4.48 ms and 9.46 ms on the DGX H100 and DGX A100, respectively. (c), (f) fix the buffer size at 4 GiB and vary the straggler delay.

## 4.1 BENCHMARKING STRAGGLAR ON MULTI-GPU SCALE-UP DOMAINS

**Setup.** We run experiments on several multi-GPU testbeds: (1) NVIDIA DGX H100, 8×80GB H100 GPUs connected by NVLink 4.0 (450 GB/s P2P) and NVSwitch 3.0; (2) NVIDIA DGX A100, 8×80GB A100 GPUs connected by NVLink 3.0 (300 GB/s P2P) and NVSwitch 2.0; (3) a node of the Perlmutter supercomputer (NERSC, 2025), with 4×40GB A100 GPUs fully connected (mesh) by NVLink 3.0 for 100 GB/s P2P (results in §G). (1) and (2) are VMs obtained through RunPod (RunPod, Inc., 2025). We use CUDA events on GPU rank 0 to measure the runtime.

**Varying buffer size.** We first assume the straggler delay can mask the REDUCESCATTER precondition and measure runtime from this point to capture performance for ideal use cases of our algorithm. (For Broadcast, we assume the entire ALLREDUCE precondition has completed.) Following the standard for benchmarking ALLREDUCE performance, we evaluate the communication time as we scale the buffer size in powers of 2 (in bytes). NCCL behaves unpredictably when the P2P data transfer size (*i.e.,* chunk size) is not a multiple of 4 KiB (page size), so we pad buffers for StragglAR to ensure the chunk size is the lowest multiple of 4 KiB *greater* than $\frac{s}{n-1}$ (for baselines, the chunking scheme inherently ensures this when $s$ is a power of 2). For each buffer size, we run 50 iterations per algorithm and report the mean, with error bars for the standard error of the mean.

Similar to how NCCL reports performance (NVIDIA, 2024b), we show the *algorithmic bandwidth*— buffer size divided by collective communication time (*i.e.,* runtime normalized by input size)—in Fig. 5 (a,d). For some smaller buffer sizes, RHD, Broadcast, and MSCCL outperform StragglAR, which still outperforms Ring; this matches expected results from $\alpha-\beta$ costs since $\alpha$ costs dominate for small buffers. However, StragglAR is consistently the fastest for the larger buffer sizes (which scale with model size), specifically 8.3% faster on the 4-A100 Perlmutter node and >25% faster on the 8-H100 and 8-A100 DGX servers. The outlier at 256 MiB likely stems from NCCL's internal tuning in the 64-512 MiB range, where internal protocol changes cause unexpected performance, as confirmed by our own `nccl-tests` profiling (§H) and prior work (Xu et al., 2025; Hu et al., 2025).

To simulate stragglers, we idle GPU rank $n-1$ for the specified number of clock cycles. Simultaneously, we begin the REDUCESCATTER for the other $n-1$ GPUs for StragglAR (similarly, ALLREDUCE for Broadcast). We conduct similar experiments as above, but instead impose the environment-specific average delay of 4.48 ms and 9.46 ms observed across our Llama-3.2-3B fine-tuning experiments on the DGX H100 and DGX A100, respectively. Fig. 5 (b,e) shows that StragglAR

improves the runtime under typical straggler delays in distributed ML with larger buffer sizes, with average-case performance closely matching ideal performance. For buffers sizes greater than 1 GiB, StragglAR's performance declines slightly from the ideal case because the REDUCESCATTER for this buffer size cannot be fully overlapped with the average straggler delay.

**Varying straggler delay.** We vary the straggler delay by adjusting the number of clock cycles for the sleep kernel, and run all algorithms on the largest buffer size (4 GiB) as a stress-test of the precondition overlap. Fig. 5 (c,f) captures the total time from when the ALLREDUCE is called on non-straggler GPUs to when it completes. It shows the critical delays of 5.53 ms and 7.57 ms for the DGX H100 and A100, respectively, after which StragglAR outperforms baselines. The critical delay is higher for the DGX A100 since a REDUCESCATTER takes longer due to its lower P2P bandwidth. The critical delay also depends on buffer size, as shown in Figs. 6a and 6b, which plot the runtime of the REDUCESCATTER precondition on the DGX H100 and A100, respectively. While the shaded regions in Figs. 6a and 6b capture the ideal straggler delays at which StragglAR achieves its full theoretical gains, StragglAR still outperforms baselines when some, but not all, of the REDUCESCATTER is overlapped with the straggler delay (see §B). For example, the critical delay for a 4 GiB buffer in Fig. 5c is less than the REDUCESCATTER time for 4 GiB shown in Fig. 6a. Thus, StragglAR enables speedups even if the straggler delay is less than the REDUCESCATTER precondition, but longer than the critical delay. As we show in §4.3 and discuss in §B, **the critical delay approaches zero as the size of the GPU cluster increases** due to StragglAR's efficient scaling.

## 4.2  END-TO-END EVALUATION ON ML WORKLOADS

We run experiments on DGX A100 VMs to assess StragglAR's impact on end-to-end training time with data-parallel fine-tuning of three popular LLMs: Llama-3.2-3B (Grattafiori et al., 2024), Phi-3-mini-3.8B (Abdin et al., 2024), and Qwen-2.5-3B (Qwen, 2024). These models are the largest that could fit in GPU memory in our hardware setup for training while maintaining a reasonable batch size, are heavily downloaded on Hugging Face (Wolf et al., 2020), and provide diversity across core open-source LLM providers (Meta, Microsoft, and Alibaba).

| Model | Speedup (%) | Straggler persistence (%) | GPU hrs. saved/day |
|---|---|---|---|
| Llama-3.2-3B | 4.75 | 90 | 9.12 |
| Phi-3-mini-3.8B | 4.43 | 95 | 8.51 |
| Qwen-2.5-3B | 2.39 | 77 | 4.59 |

Table 2: End-to-end training speedups over Ring, and GPU-hours saved/day on an 8-GPU VM. Values reflect worst-case speedups on the given VM due to static straggler detection (dynamic detection could enable further speedups).

We compile our ALLREDUCE implementations into a custom package and configure PyTorch to call this backend. In each VM, we first profile the workload with standard PyTorch tools and identify persistent stragglers, *i.e.,* ranks that are most likely to be delayed based on data from prior runs. We use this approach both because persistent stragglers are common according to prior work (Lin et al., 2025; Jiang et al., 2024) and our own experiments (§C), and because it stress-tests StragglAR to encounter both ideal and worst-case conditions (when a different/no rank is the straggler). Then, we pass the selected rank to the backend and train the model for 100 iterations (batch size of 32). Because the buffer sizes of these models are $\geq$ 1 GiB, we compare StragglAR to the Ring algorithm, which is optimal at large buffer sizes (Fig. 5a). We show the end-to-end speedups in Table 2.

The end-to-end speedup depends on several factors: how often the actual straggler was the rank we passed to the backend, the straggler's delay, and the fraction of overall time spent on ALLREDUCE. For Qwen-2.5-3B, gains are smaller because the straggler on that VM was less persistent, leading our algorithm to encounter its worst-case scenario (*i.e.,* no REDUCESCATTER overlap) more often. Even then, StragglAR consistently shows end-to-end gains, as its worst-case performance nearly matches baselines while its upside is much higher (Fig. 2b). StragglAR does not require online straggler detection with dynamic stragglers, as eager conditional execution of schedules based on the first $n-1$ ready ranks means at worst (*i.e.,* no straggler delay), StragglAR's performance closely matches baselines. These speedups translate to 9.12 GPU-hours saved per day on an 8-GPU server (Table 2).

## 4.3  SCALING CHARACTERISTICS OF STRAGGLAR

To assess scaling to the largest practical scale-up domains (256 GPUs), we employ the same approach as prior work: using the popular and empirically validated analytical network model to simulate

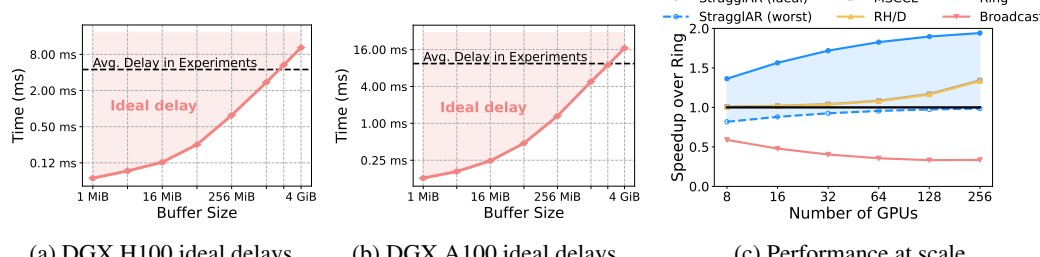

(a) DGX H100 ideal delays.  (b) DGX A100 ideal delays.  (c) Performance at scale.

Figure 6: (a), (b) REDUCESCATTER time over buffer sizes, capturing the range of straggler delays where StragglAR achieves full theoretical guarantees for the DGX H100 and A100, respectively. (c) ALLREDUCE performance scaling with the analytical model for a 1 GiB buffer. The shaded region is the range of StragglAR's possible speedups over the Ring algorithm.

performance on clusters larger than 8 GPUs (Won et al., 2023; Wang et al., 2025; Gui et al., 2025; Won et al., 2024), as we lack access to hardware like NVIDIA's GB200 (NVIDIA, 2024). The analytical simulator uses the $\alpha-\beta$ model to predict performance. We use $\alpha = 3\mu s$ based on latency profiling studies of recent NVLinks (Microway, 2024) and $\beta$ as the inverse of 450 GB/s, the P2P bandwidth on DGX H100. In these simulations, we capture the *entire range* of performance supported by StragglAR, from the worst case where none of the initial REDUCESCATTER can be overlapped to the ideal case where the straggler delay enables full overlap (see Figs. 6a, 6b for ideal range). In Fig. 6c, both StragglAR's ideal and worst-case performance improve with cluster size for realistic, large buffer sizes (1 GiB) in distributed ML. By $n = 256$, StragglAR provides a nearly $2\times$ speedup over the Ring algorithm in straggler settings while being *no worse* even without stragglers. Thus, incorrect or infeasible straggler detection has minimal impact, as StragglAR exhibits competitive bandwidth efficiency even in worst-case conditions (no straggler delay). As shown in §B, the critical delay—straggler delay required for StragglAR to outperform Ring—*decreases* as the cluster size ($n$) *increases* for any given buffer size ($s$) and inter-GPU link bandwidth ($1/\beta$), due to StragglAR's efficient scaling. We report end-to-end scaling results in simulation for ML training in §J.

**Limitations.** To consistently achieve its best-case performance with dynamic stragglers, StragglAR requires conditional execution of schedules based on the $n-1$ ranks that are first ready, which can be complex. Further, StragglAR relies on two synchronization barriers, which adds implementation complexity. Although the additional barrier introduces overhead, our experiments in §4 indicate that this overhead is minimal compared to StragglAR's performance gains.

While StragglAR performs on par with baselines at scale even without a straggler, its performance on smaller clusters depends on the critical delay, which varies with the exact GPU P2P bandwidth. Our algorithm also does not support odd values of $n$, though such setups are atypical in large-scale ML. While StragglAR can tolerate multiple stragglers, since a straggler is by definition relative, it is less effective when many GPUs straggle *simultaneously*; however, this scenario is highly improbable since GPU execution times are continuous variables (Dean and Barroso, 2013; Warraich et al., 2025). Further, there may be settings in which the data transmission overhead is so high that synchronization overheads (*e.g.,* stragglers) are less relevant for optimizing communication (*e.g.,* very low link bandwidth). While StragglAR may not provide significant speedups in these settings—since we target use cases with stragglers—its bandwidth efficiency, even in the worst-case scenario with no straggler delay, makes it a competitive and generic ALLREDUCE algorithm regardless of stragglers. Finally, *completely asynchronous* techniques that drop the straggler's data (Recht et al., 2011) may be preferable for specific applications with high error tolerance or if the straggler delay is very severe.

## 5 CONCLUSION

We design StragglAR, a parallel algorithm that exploits natural variation in GPU execution times to speed up ALLREDUCE. StragglAR achieves a $2\times$ speedup over the known lower bound for bandwidth-optimal ALLREDUCE, while performing as well in the worst case. By introducing the dimension of *temporal asymmetry*, StragglAR offers a new paradigm for collective algorithm design.

## REPRODUCIBILITY STATEMENT

We detail the assumptions, theoretical guarantees, and experimental procedures required to reproduce our results. The formal description of StragglAR and all assumptions appear in § 3, while the complete proofs are provided in the Appendix (§D). These proofs mirror the intuition in the main paper and explicitly reference every prerequisite lemma.

Our implementation consists of two components: the offline schedule generator and the NCCL-based runtime. § 4 enumerates the exact API calls we use (`ncclSend()`, `ncclRecv()`, `ncclReduceScatter()`), describes the synchronization barriers, and specifies how the straggler is emulated for benchmarking experiments and how it is detected natively for end-to-end experiments. The same section lists every baseline, including algorithmic variants and hyperparameters, so that readers can rebuild the comparisons without relying on vendor implementations. Hardware configurations (GPU model, interconnect, bandwidth, and memory) are also documented there. We also discuss the parallelization strategy, batch sizes, and open-source models used for the motivational experiments (Fig. 2a and §C) and end-to-end evaluation (§4).

The supplementary archive submitted with this work contains the source code for both components, instructions for compiling the CUDA kernels, a driver to launch the experiments reported in Figures 5–6, and configuration files that encode the straggler delays and buffer sizes we sweep. The scripts emit raw runtime traces and summary tables identical to those used to generate the plots in the paper.

We do not release new datasets or pretrained models. Our work operates purely on synthetic buffers and standard collective-communication APIs. The limitations of StragglAR are discussed in §3 and revisited in § 4. Together, these details satisfy the ICLR 2026 reproducibility guidelines.

## LLM USAGE

We used LLMs to help us identify writing errors, such as spelling and grammar mistakes, and to help debug scripts for plotting figures.

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

APPENDIX

# A    EXTENDED RELATED WORK

We provide more discussion of related work in this section.

In distributed training and inference, each GPU produces local gradients and activations that must be aggregated across devices. This multi-device aggregation is performed using the ALLREDUCE collective communication primitive, which transmits and reduces data across the inter-GPU network using optimized parallel communication algorithms. The parallelization strategy of an ML job determines which GPUs communicate and how often: data parallelism uses ALLREDUCE regularly to average gradients, while tensor model parallelism (Shoeybi et al., 2019a) invokes ALLREDUCE many times within each model pass to exchange activations. In ML software stacks, collective communication libraries (CCLs) like NCCL implement algorithms for collective primitives, interfacing with ML application libraries (*e.g.,* PyTorch (PyTorch Contributors, 2024), TensorFlow (TensorFlow Developers, 2024)) at one end and network transport protocols at the other. For ALLREDUCE, NCCL chooses between multiple algorithms based on the number of GPUs and the amount of data to be reduced. Since buffer sizes in modern ML workloads tend to be large (Shah et al., 2023), CCLs leverage bandwidth-optimal algorithms (*e.g.,* Ring) to minimize the ALLREDUCE time.

Most CCLs, including NCCL, adopt a bulk-synchronous model where all GPUs synchronize before the collective, guaranteeing data correctness and enabling efficient parallel communication algorithms that rely on all ranks to implement the collective concurrently. However, performance degrades significantly when one or more GPUs are slow to reach the synchronization point. The slowest GPU (the straggler) dictates overall performance due to tail effects (Warraich et al., 2025; Dean and Barroso, 2013; Wang et al., 2024a; Gangidi et al., 2024; Li et al., 2014). Even with multiple slower GPUs, there is typically only a single straggler since the probability of multiple GPUs completing exactly simultaneously is extremely small. Straggler delays may stem from hardware issues like thermal throttling and power supply, or runtime factors like network congestion, background noise, and unavoidable skewed compute allocations across GPUs (Wu et al., 2024; Jiang et al., 2024; Grattafiori et al., 2024; Xiong et al., 2024). Recent work has highlighted the significant impact of stragglers on datacenter-scale ML jobs (Jiang et al., 2024; Lin et al., 2025), and our experiments fine-tuning Llama-3.2 (1B/3B) across different hardware setups (Fig. 2a) reveal straggler effects of up to 30 ms even within individual multi-GPU servers.

| Method | Domain | Backend | Application | Effect on Loss/Convergence |
|---|---|---|---|---|
| (Harlap et al., 2016) | Scale-out | Custom C++ runtime | DP | No |
| (Karakus et al., 2017) | Scale-out | MPI | DP | Yes |
| (Wang et al., 2020) | Scale-out | NCCL | DP, TP | No |
| (Won et al., 2024) | Irregular scale-up | Simulation | DP, TP | No |
| (Zhao et al., 2024a) | Scale-out | Custom CUDA runtime | DP, TP, EP | No |
| (Warraich et al., 2025) | Scale-out | Gloo & custom transport | DP | Yes |
| **StragglAR (Ours)** | **Scale-up** | **NCCL** | **DP, TP** | **No** |

Table 3: StragglAR vs. prior work (DP=data parallel, TP=tensor parallel, EP=expert parallel). While StragglAR targets the scale-up domain, it also applies to switched homogeneous scale-out domains, like in today's rail-optimized topologies for ML.

**Straggler mitigation.** Some prior works attempt to identify and remove straggling devices or thoroughly investigate root causes (Jiang et al., 2024; Lin et al., 2025). However, removing stragglers wastes compute capacity, and only applies to severe stragglers rather than the routine compute heterogeneity we observe regularly, even at small scales. Moreover, stragglers arise from many sources, making diagnosis difficult and incomplete (Lin et al., 2025). Recent works obviate straggler delays by approximating or dropping the straggler's data during training (Warraich et al., 2025; Harlap et al., 2016; Karakus et al., 2017; Recht et al., 2011). While these approximations and asynchronous techniques can progress without waiting for stragglers, they are limited to data-parallel training and can impact model convergence. In contrast, StragglAR is a fundamentally new ALLREDUCE algorithm that ensures accurate, unmodified reductions and is agnostic to the ALLREDUCE use case, making it applicable to both data-parallel training and tensor-parallel training/inference. Like all collective algorithms, StragglAR requires participation from all GPUs (including the straggler), but accelerates the collective itself. Other works mitigate stragglers in datacenter-scale ML jobs through

systems-level workload rebalancing and runtime strategies (Wu et al., 2024; Zhao et al., 2024a), improving efficiency by adapting task placement or dynamically selecting among different known collective algorithms based on profiling. For example, our investigation of AdapCC (Zhao et al., 2024a) finds that it relies classical Tree algorithms, which sacrifice bandwidth to improve latency. In contrast, StragglAR is not a runtime controller, but a new bandwidth-efficient parallel algorithm for ALLREDUCE that changes the communication schedule itself to provably reduce communication complexity. It can be integrated with different runtime systems that implement these algorithms.

**ALLREDUCE algorithms.** Collective algorithms are designed based on knowledge of the underlying network topology. Inter-GPU networks typically fall into two categories: scale-up (homogeneous high-bandwidth links within a node or rack) and scale-out (heterogeneous or unpredictable bandwidth across multiple nodes or racks). Recent works have synthesized new ALLREDUCE algorithms for heterogeneous networks with asymmetric bandwidth (Shah et al., 2023; Zhao et al., 2024a; Wang et al., 2020; Won et al., 2024; Zhang et al., 2024). However, for the homogeneous networks that define modern scale-up domains, these synthesizers invariably converge to bandwidth-optimal classical algorithms, such as Ring and Recursive Halving/Doubling (Shah et al., 2023; Wang et al., 2020). Consequently, state-of-the-art CCLs like NCCL continue to implement classical algorithms for collective communication in scale-up domains (Hu et al., 2025). We provide Table 3 to concretely situate our contribution in the space of collective algorithms and straggler mitigation strategies. We note that while we focus on the scale-up domain for StragglAR, it applies to any homogeneous switched network, including rail-optimized scale-out domains (NVIDIA, 2022).

**Optimal collective communication.** Over decades, researchers have implemented collective communication primitives with fast algorithms that are carefully attuned to latency and bandwidth trade-offs. With today's large ML models that require communicating large amounts of data, bandwidth-efficient algorithms are of primary focus (Narayanan et al., 2021; Fei et al., 2021). In homogeneous networks, well-known parallel algorithms, such as Ring, recursive halving/doubling, *etc.*, enable fast, bandwidth-efficient ALLREDUCE by heavily parallelizing communication in every round of the algorithm (Patarasuk and Yuan, 2009; Rabenseifner, 2004; Rabenseifner and Träff, 2004; Thakur et al., 2005; Bruck et al., 1994; Sanders et al., 2009). For heterogeneous networks, recent works synthesize custom collective algorithms by using the hardware topology or parallelization strategy as an input (Shah et al., 2023; Kim et al., 2024; Cai et al., 2021; Zhao et al., 2024b; Laskar et al., 2024; Won et al., 2024; Mahajan et al., 2023); optimizing a collective algorithm to minimize the latency and bandwidth cost, however, is a challenging combinatorial optimization problem that often requires hand-tuned heuristics to constrain the search space (Shah et al., 2023). Further, some approaches optimize ALLREDUCE for a specific topology (Jain and Sabharwal, 2010; De Sensi et al., 2024; Wang et al., 2023a). Our work extends this line of research by optimizing ALLREDUCE in settings with a straggler for distributed ML, where bandwidth-efficient algorithms are required due to the large buffer sizes (Narayanan et al., 2021; Shah et al., 2023; NVIDIA, 2024a).

**Rebalancing workloads.** Recent works improve the efficiency of distributed ML by optimizing the model parallelization strategy, which can affect data/work sharding across GPUs and the collective communication incurred (Jia et al., 2019; Zhang et al., 2024; Zheng et al., 2022; Shoeybi et al., 2019b; Rajbhandari et al., 2020; Grattafiori et al., 2024; Narayanan et al., 2021; Unger et al., 2022; Zhao et al., 2024a). Vanilla parallelization strategies evenly shard or replicate the compute and memory load (*i.e.,* model weights, input batches, activations, optimizer states, *etc.*) across GPUs within a group to achieve speedups from parallelization. This balances the computation and communication load experienced by each worker, but could introduce resource idling when there are slow GPUs in the group. Our experiments in Fig. 8 and §C highlight significant straggler effects even with uniform workload placement. Other works identify severe stragglers and balance the workload dynamically to limit their impact (Harlap et al., 2016; Wu et al., 2024; Wang et al., 2023b; Lin et al., 2025). StragglAR directly addresses stragglers at a lower layer of abstraction, within the ALLREDUCE kernel, and thus expands on the problems tackled by these works.

**Sending less data.** Various methods leverage the resilience of stochastic optimization algorithms used in training to reduce the communication volume by selectively communicating data with high statistical value (Warraich et al., 2025; Li and Hoefler, 2022; Wang et al., 2023c), synchronizing gradients with varying frequencies (Giladi et al., 2023; Wang and Varma, 2024; Liu et al., 2022; Karakus et al., 2017; Yang et al., 2020), or sending lower-precision or compressed data (Alistarh et al., 2017; Bernstein et al., 2018; Wang et al., 2022a; 2024b; 2023c; Lu et al., 2022; Tang et al., 2021; Renggli et al., 2019; M Abdelmoniem et al., 2021; Wang et al., 2023d). StragglAR is

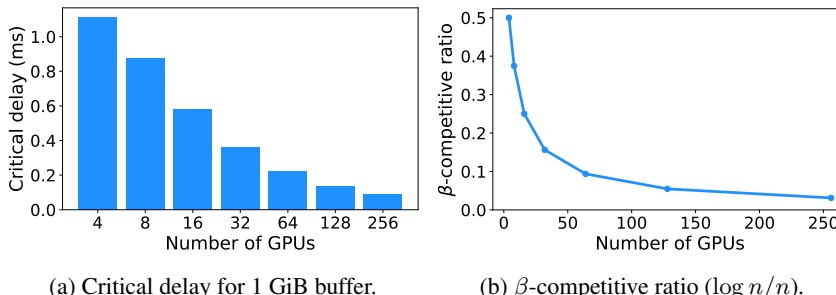

(a) Critical delay for 1 GiB buffer.      (b) $\beta$-competitive ratio ($\log n / n$).

Figure 7: Scaling simulations of StragglAR's critical delay. (a) shows the critical delay in ms for a 1 GiB buffer as cluster size increases. (b) shows the $\beta$-competitive ratio, independent of the buffer size.

orthogonal to these works because it optimizes the collective algorithm itself (*i.e., exactly how* the data is transmitted as opposed to *how much*) and can operate given any buffer size, including after compression. Further, because StragglAR directly modifies the collective algorithm, as opposed to the data that is sent or the frequency of synchronization, it also supports ALLREDUCE for tensor parallelism, which is used in both training and inference (Shoeybi et al., 2019a).

**Compute-communication overlap.** Recent works propose fine-grained overlap of compute and communication to mitigate synchronization delays and reduce communication overheads in distributed ML (Ismayilov et al., 2023; Jangda et al., 2022; Wang et al., 2023e; NVIDIA, 2024c; Punniyamurthy et al., 2024; Liang et al., 2024; Pati et al., 2024; Chen et al., 2024; Wang et al., 2022b). Although this approach may be useful for small, irregular communication patterns, such as expert-parallel ALLTOALL (Liu et al., 2024), it is less useful with large data buffers that require aggregation and are bottlenecked by network bandwidth (Dryden et al., 2018; Patarasuk and Yuan, 2009). Thus, bulk data transfers, like gradient synchronization in data-parallel training or activation aggregation in tensor-parallel training with large batch sizes, rely on bandwidth-optimal collective communication. Further, these works consider overlapping compute and communication at higher layers of abstraction, but rely on the same algorithms to implement the collective communication. StragglAR instead brings the idea of overlap to the communication algorithm itself, and uses this to directly address the straggler problem. Recent research into CCLs that provide finer-grained control over communication (Shah et al., 2025) could make it easier to implement and execute overlapped algorithms like ours. Other works propose entirely abandoning synchronization barriers and relying on resilience of stochastic gradient descent to enable convergence during training (Nabli et al., 2023; Tyurin et al., 2024; Su et al., 2022; Tyurin and Richtarik, 2024; De Vos et al., 2023; Recht et al., 2011). However, these approaches impact model convergence, limiting applicability, and do not generalize to tensor-parallel training and inference, which require accurate and complete reductions.

# B  STRAGGLER DELAY REQUIREMENTS

Aside from the standard parameters that dictate the performance of any collective communication algorithm (*e.g.,* $\alpha$, $\beta$, $s$, and $n$), StragglAR's performance is also affected by the straggler delay. With no straggler delay, the initial REDUCESCATTER cannot be overlapped at all and StragglAR achieves its worst-case performance. With "ideal" straggler delay that exceeds the REDUCESCATTER time, the initial REDUCESCATTER can be fully overlapped and StragglAR achieves its ideal performance. A straggler delay between these two values will cause StragglAR to achieve intermediate performance between the ideal and worst-case bounds.

There are two important straggler delay thresholds that help quantify performance: (1) *ideal delay*, which is the straggler delay required to achieve the ideal-case bound, and (2) *critical delay*, which is the straggler delay required to ensure that StragglAR does not underperform the Ring algorithm. The expression for the ideal delay (1) is simply the time to complete the REDUCESCATTER among $n-1$ ranks: $T_{RS} = (n-2)\alpha + \frac{n-2}{n-1}s\beta$. We denote the random variable for the straggler delay as $T_{straggler}$. As long as $T_{straggler} \geq T_{RS}$, the ideal-case performance bound is achieved. $T_{RS}$ inherently depends on the number of GPUs ($n$), the hardware environment—which determines the values of $\alpha$ (*i.e.,* link latency and software overheads) and $\beta$ (*i.e.,* inverse of link bandwidth)—and the buffer size ($s$). In Figs. 6a and 6b, we empirically show the value of $T_{RS}$ on the DGX H100 and A100, respectively, for

different buffer sizes by directly running the REDUCESCATTER on the hardware. Fig. 6b shows a similar pattern as Fig. 6a (ideal straggler delay for DGX H100) with buffer size, but the raw values are proportionally higher in Fig. 6b because the DGX A100 has a lower P2P bandwidth of 300 GB/s compared to 450 GB/s P2P bandwidth on the DGX H100. However, straggler delays also tend to be substantially higher on A100s, as shown by comparing the *Avg. Delay in Experiments* line in each figure, such that the average empirical delay is still higher than the ideal delay for all buffer sizes besides the largest, 4 GiB. This suggests StragglAR can reliably achieve its ideal performance in this setup with buffer sizes up to 2 GiB. For even larger buffer sizes, performance will fall in between the ideal and worst-case bounds, since the REDUCESCATTER cannot be fully overlapped with the average straggler delay. The average straggler delay will also likely vary for different workloads.

To derive the *critical delay*, we solve the inequality below for $T_{straggler}$, the straggler delay:

$$T_{RS} + T_{SAR} \leq T_{straggler} + T_{Ring}$$

$$(n-2)\alpha + \tfrac{n-2}{n-1}s\beta + (n-2+\log n)\alpha + \tfrac{n-2+\log n}{n-1}s\beta \leq T_{straggler} + 2(n-1)\alpha + \tfrac{2(n-1)}{n}s\beta$$

$$T_{straggler} \geq \big(2(n-2)+\log n\big)\alpha + \tfrac{2(n-2)+\log n}{n-1}s\beta - 2(n-1)\alpha - \tfrac{2(n-1)}{n}s\beta$$

Since $\tfrac{2(n-2)+\log n}{n-1}s\beta \approx \tfrac{2(n-1)+\log n}{n}s\beta$, we simplify the above expression as

$$T_{straggler} \geq \big(2(n-2)+\log n\big)\alpha + \tfrac{2(n-1)+\log n}{n}s\beta - 2(n-1)\alpha - \tfrac{2(n-1)}{n}s\beta$$

$$T_{straggler} \geq (\log n - 2)\alpha + \tfrac{\log n}{n}s\beta$$

If $T_{straggler} > (\log n - 2)\alpha + \tfrac{\log n}{n}s\beta$, StragglAR outperforms the Ring algorithm. If $T_{straggler}$ equals this expression, StragglAR performs on par with Ring and if $T_{straggler}$ is less than this expression, StragglAR underperforms Ring. Using the methodology from §4.3, we show the critical delay for a 1 GiB buffer as the size of the GPU cluster ($n$) increases in Fig. 7a. The critical delay *decreases* as the number of GPUs in the cluster increases, and is less than 0.1 ms for $n$=256.

With $\alpha = 3\mu s$ (as used in §4.3 (Microway, 2024)), even at the largest scale of $n$=256, the latency cost only contributes $(\log(256)-2)\alpha = 18\mu s$ (*i.e.,* 0.018 ms) regardless of buffer size. Prior work's estimate of $\alpha$=0.7$\mu s$ (Shah et al., 2023) would only lower the contribution of the latency cost even further. Due to the negligible latency cost, especially at larger buffer sizes, we focus on the bandwidth-derived component of the critical delay: $\tfrac{\log n}{n}s\beta$. We plot the *$\beta$-competitive-ratio* $\tfrac{\log n}{n}$ (*i.e.,* coefficient for the $\beta$-derived component of the critical delay) as $n$ increases in Fig. 7b. The figure shows that this ratio asymptotically approaches 0 as the number of GPUs increases, thereby explaining the decreasing trend in Fig. 7a. This means that at scale, StragglAR is a *generic* ALLREDUCE algorithm: with stragglers, it provides 2× speedups over the best known bandwidth-efficient algorithms, while still performing on par with these baselines even when there is no straggler or multiple simultaneous stragglers (*i.e.,* no straggler delay).

## C  PERSISTENT STRAGGLERS

While recent works (Jiang et al., 2024; Lin et al., 2025) have shown persistent stragglers in ML jobs with thousands of GPUs, we show that even smaller multi-GPU jobs encounter such stragglers. We run Llama-3.2 fine-tuning jobs in three hardware environments: (1) 4 A100s with 40GB memory and NVLink 3.0 fully-connected mesh network in the Perlmutter supercomputer (NERSC, 2025), (2) 4 A100s with 80GB memory in RunPod (RunPod, Inc., 2025), and (3) 8 A100s with 80GB memory in RunPod. Both RunPod servers use NVLink 3.0 with an NVSwitched any-to-any network. For every environment, we repeat the same workload three times.

We encounter stragglers in all nine runs. We define the straggler delay as $T_{delay} = T_{rest} - T_{straggler}$, where $T_{straggler}$ is the ALLREDUCE execution time of the straggler — the GPU that started the ALLREDUCE last (rank 0 in Fig. 1) — and $T_{rest}$ is the ALLREDUCE execution time of the second-slowest rank. Note that $T_{straggler}$ is the shortest ALLREDUCE time among all GPUs because the straggler spends no time waiting for others before communicating, and our measure of $T_{delay}$ is conservative since it only encodes the idle time of the second-slowest GPU. We find a median straggler delay of more than 8 ms and a mean delay of 9.45 ms. The straggler GPU rank persists not only across

iterations, but also across runs. Fig. 8 shows that one specific GPU is the slowest in a communication group in 98% of iterations, and is also the persistent straggler over multiple independent runs of the same workload on the same VM. Fig. 8(a) shows these findings also generalize to multiple stragglers. Our experiments document (1) the severity of straggler GPUs that consistently delay ALLREDUCE, and (2) that the same GPU rank is typically the straggler, allowing us to use offline profiling to bootstrap StragglAR. These findings also suggest that routine stragglers may even appear from intrinsic and unavoidable GPU hardware heterogeneity (even with the same GPU model).

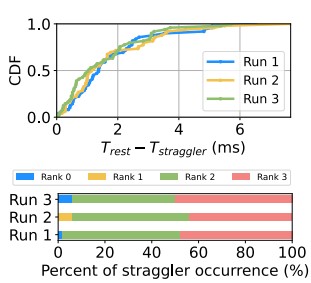 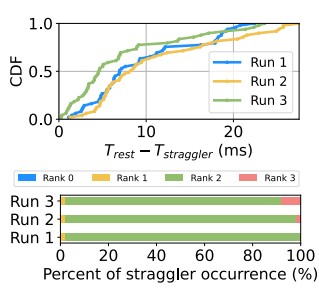 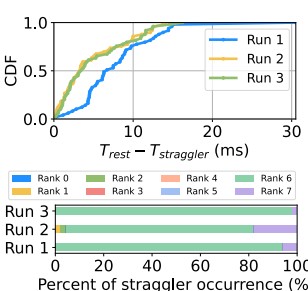

(a) Perlmutter, 4 A100 SXM, Llama-3.2-1B, batch_size=32.

(b) RunPod, 4 A100 SXM, Llama-3.2-3B, batch_size=32.

(c) RunPod, 8 A100 SXM, Llama-3.2-3B, batch_size=64.

Figure 8: CDFs of the straggler delay, $T_{rest} - T_{straggler}$, and distribution of straggler ranks.

# D PROOF OF COMMUNICATION COMPLEXITY

First, we prove that schedules synthesized by StragglAR for power-of-2 $n$ are guaranteed to complete within $n + \log n - 2$ rounds. As in Algorithm 1, $A$ represents the data structure that maps each active chunk to the set of non-straggler ranks that holds it (and is updated in each round). We assume this data structure is a hash table for constant-time updates and access. Note that an *active chunk* necessarily implies that the chunk has been fully reduced. Our code generates results that match the proof bounds.

**Lemma 1.** *Prior to round* $\log n$, *every non-straggler rank possesses exactly one active chunk such that for* $j = 0, \ldots, \log n - 1, |A[c_j]| = 2^{\log n - 1 - j}$.

*Proof.* After round $j < \log n$ has completed, $|A[c_j]| = 1$ because $c_j$ has just been fully reduced (via a pairing between rank $j$ and the straggler), and only rank $j$ holds it. In round $j + 1$, rank $j$ sends $c_j$ to rank $j + \log n$, so $|A[c_j]| = 2$. In each subsequent round $r = j + i$ for $i < \log n - j$, all ranks in $A[c_j]$ send $c_j$ to distinct ranks lacking a chunk, thus doubling $|A[c_j]|$ before the next round $r + 1$.

Therefore, the number of non-straggler ranks possessing $c_j$ after round $r$ for $j \leq r < \log n$ is $|A[c_j]| = 2^{r-j}$. After any round $r < \log n$, the total number of non-straggler ranks holding an active chunk is

$$\sum_{j=0}^{r} 2^{r-j}$$

We show that there are enough non-stragglers without a chunk after each round $r < \log n - 2$ (equivalently, before each round $r < \log n - 1$) to ensure that every rank with an active chunk has a distinct target it can send to:

$$\sum_{j=0}^{r} 2^{r-j} \leq (n-2) - \sum_{j=0}^{r} 2^{r-j}$$

($n-2$ appears due to the straggler pairing in round $r$). Substituting the closed form of the sum,

$$2^{r+1} - 1 \leq (n-2) - (2^{r+1} - 1) = n - 2^{r+1} - 1,$$

which simplifies to

$$2^{r+2} \leq n.$$

This inequality is only violated when $r+2 > \log n$. Thus, after each round $r \le \log n - 2$ (before each round $r \le \log n - 1$) there are strictly more chunk-free non-stragglers than senders, so every sender can choose a distinct receiver. After round $r = \log n - 1$ (*i.e.,* last round of Phase 1), $|A[c_j]| = 2^{\log n - 1 - j}$, so the total number of non-straggler ranks with an active chunk is given by

$$\sum_{j=0}^{\log n - 1} 2^{\log n - 1 - j} = \sum_{i=0}^{\log n - 1} 2^i = 2^{\log n} - 1 = n - 1$$

This means all $n-1$ non-straggler ranks have a single fully reduced chunk after Phase 1. $\square$

**Lemma 2.** *Each chunk $c_r$, $r < n-1$, is fully reduced in round $r$, and is fully propagated to all ranks immediately after round $r + \log n$.*

*Proof.* The first part of this statement follows directly from the schedule, as it dictates that partially reduced chunk $c_r$ resides on rank $r$ given the REDUCESCATTER precondition, and rank $r$ communicates with the straggler, $\sigma$, in round $r$, enabling $c_r$ to be fully reduced on ranks $r$ and $\sigma$.

To prove the second part of this statement, we first inductively prove the following invariant.

Invariant $\mathcal{I}(r)$ holds immediately *prior* to round $r \ge \log n$:

(1) $|A[c_j]| = 2^{r-j-1}$, $\quad j = r - \log n, \ldots, r-1$

(2) $P_r = A[c_{r-\log n}]$ with $|P_r| = \frac{n}{2}$ and $r \in P_r$;

(3) $Q_r = \bigcup_{j=1}^{\log n - 1} A[c_{r-j}]$ with $|Q_r| = \frac{n}{2} - 1$;

(4) Active chunks are pairwise disjoint: $A[c_i] \cap A[c_j] = \varnothing, i \ne j, \forall i, j \in \{r - \log n, \ldots, r-1\}$.

This enables us to match $u \in P_r \setminus \{r\}$ with $v \in Q_r$ to ensure that chunk $c_r$ has been fully propagated to all ranks immediately after round $r + \log n$.

**Base case.** We rely on Lemma 1 for the base case. Immediately before round $\log n$ (after round $\log n - 1$), $|A[c_j]| = 2^{\log n - 1 - j}$, where sets $A[c_j]$ are pairwise disjoint for all $j = 0, 1, \ldots, \log n - 1$. Thus, $|P_{\log n}| = |A_{c_0}| = 2^{\log n - 1} = \frac{n}{2}$ and $|Q_{\log n}| = n - 1 - \frac{n}{2} = \frac{n}{2} - 1$. Since rank $\log n$ holds chunk $c_0$, as the protocol in Algorithm 1 dictates that rank 0 sends $c_0$ to rank $\log n$ in round 1, we know $\log n \in P_{\log n}$. By matching $u \in P_{\log n} \setminus \{\log n\}$ with $v \in Q_{\log n}$ and having $u$ and $v$ each send their active chunks, we ensure that $c_0$ has fully propagated after round $r = \log n$. A matching is possible because every rank in $P_{\log n}$ lacks the active chunk of every rank in $Q_{\log n}$, and vice versa. After the round, $c_0$ has fully propagated and every other active chunk has doubled.

**Inductive step.** Assume $\mathcal{I}(r)$ holds immediately before round $r$ and that in round $r$ we match some $u \in P_r \setminus \{r\}$ with a distinct $v \in Q_r$. Because $u$ sends $c_{r-\log n}$ to $v$, we have:

1. *Full propagation.* After round $r$, every non-straggler now holds $c_{r-\log n}$ (since all ranks in $P_r \bigcup Q_r$ now have it), so this chunk is no longer active.

2. *Doubling of the remaining active chunks.* Every other active chunk is sent and received exactly once, so its cardinality doubles; every rank still stores exactly one active chunk. If a rank possessed the active chunk that just expired, it received a new one, allowing it to maintain having exactly one active chunk. If a rank possessed another active chunk, it received the active chunk that just expired, allowing it to maintain the same status.

We verify the validity of $\mathcal{I}(r+1)$ before round $r+1$ as follows. We can remove $c_{r-\log n}$ from $A$ because it has fully propagated via the matching between $P_r$ and $Q_r$ in round $r$. We add $c_r$ to $A$ such that $A[c_r] = \{r\}$, since it has just been fully reduced in round $r$ via the straggler pairing. Every other active chunk has doubled in propagation (*i.e.,* the number of ranks that possess it) via the perfect matching between $P_r$ and $Q_r$. Let $j' = r + 1 - \log n$. Immediately *before* round $r+1$ we verify:

(1) For every $k \in \{j', \ldots, r-1\}$, $|A[c_k]| = 2|A[c_k]|_{(\text{before round } r)} = 2(2^{r-k-1}) = 2^{r-k}$

(2) By definition $P_{r+1} = A[c_{j'}]$. Via (1), we have $|P_{r+1}| = 2^{\log n - 1} = \frac{n}{2}$. We prove that $r+1 \in P_{r+1}$ separately below.

(3) The cardinality of the remaining $\log n - 1$ active chunks ensures the $Q_{r+1}$ constraint:

$$|Q_{r+1}| = \sum_{k=j'+1}^{r} |A[c_k]| = \sum_{t=1}^{\log n - 1} 2^{\log n - 1 - t} = \frac{n}{2} - 1$$

(4) Since each rank still holds exactly one active chunk, the sets $A[c_k]$ remain pairwise disjoint. In round $r$, ranks in $Q_r$ received $c_{r-\log n}$, which is no longer active by round $r+1$, so these ranks still retain the same active chunk they had prior to round $r$. Ranks in $P_r \setminus \{r\}$ had only active chunk $c_{r-\log n}$ before round $r$, which expired as an active chunk, and received some other active chunk in round $r$, so they also only have one active chunk each. Finally, $r \in P_r$, so its active chunk expired after round $r$, and it received active chunk $c_r$ via the straggler pairing, so it also only has one active chunk. This ensures pairwise disjointness.

There are two remaining components of the proof for the inductive step: (1) that $r+1 \in P_{r+1}$, and (2) that there always exists a feasible matching between $P_{r+1} \setminus \{r+1\}$ and $Q_{r+1}$. First, it is guaranteed that $r+1 \in P_{r+1}$ because of the rule for matching ranks in the critical window from Algorithm 1. Specifically, we ensure that rank $r+1$ is never paired with a rank whose active chunk is $c_l$ for $l > r+1-\log n$. This rule ensures that $r+1 \in P_{r+1}$ because the only active chunk it could have at the beginning of round $r+1$ is then $c_{r+1-\log n}$, guaranteeing that $r+1 \in P_{r+1}$.

The next question is whether we can ensure that there exists a matching in round $r+1$. First, we need to ensure that any rank $l$ in the critical window is not matched with a rank whose active chunk will continue to be active by round $l$. If $l \in Q_{r+1}$, this is trivial because the rank does not receive any new active chunk in that round (since it only receives the chunk that is due in that round). If $l \in P_{r+1}$, we just have to ensure that $l$ is not matched with any of the "newest" active chunks; in fact, matching $l$ with ranks possessing the second-oldest active chunk always works to keep $l$ in future $P_r$. Since $G_r$, the graph that connects $P_r$ and $Q_r$, is a complete bipartite graph, we can easily guarantee this pairing for any $l$. By removing these $l$ and their partners from their respective sets (either $P_r$ or $Q_r$), we can trivially apply Hall's Marriage Theorem since $|P_r| = |Q_r|$ (and $G_r$ is bipartite) to verify that a matching exists. Hence, $\mathcal{I}(r+1)$ holds, completing the inductive step. $\quad\square$

**Remark 1** *The final chunk, $c_{n-2}$, propagates in $\log n - 1$ rounds.*

After round $n-2$, the straggler, $\sigma$, now has all fully reduced chunks, and is no longer constrained by whom to pair with. This means we can add the straggler to $Q_r$ for subsequent rounds $r \geq n-1$ such that $|P_r| = |Q_r| = \frac{n}{2}$. (We enforce the invariant for subsequent rounds that the straggler can only send $c_{n-2}$.) Both ranks $n-2$ and $\sigma$ are in $Q_r$ (and will remain in $Q_r$ until the end of the algorithm), and their active chunk is $c_{n-2}$. Thus, $c_{n-2}$ begins with 2 ranks holding it, as opposed to 1, jumpstarting its doubling process. Every chunk can continue to double because $|P_r| = |Q_r|$ and every rank only has one active chunk, so via Hall's Marriage Theorem a perfect matching exists. Therefore, only $\log n - 1$ rounds are required to fully propagate the final chunk.

**Theorem 1.** ALLREDUCE *schedules generated by StragglAR complete in $n + \log n - 2$ rounds.*

*Proof.* Naturally, $n-1$ rounds are required for all chunks to become fully reduced, via the linear sequence of pairing with the straggler, with chunk $c_j$ fully reduced in round $j$. Lemma 2 proves that each chunk $c_j$ propagates by round $j + \log n$. Thus, the total number of rounds in the algorithm is bounded by the final chunks. Chunk $c_{n-3}$ will propagate by round $n-3+\log n$ and chunk $c_{n-2}$ will have fully propagated after round $n-2+\log n - 1 = n-3+\log n$. Since we zero-index rounds, this results in $n + \log n - 2$ rounds in total. $\quad\square$

# E  STRAGGLAR FOR NON-POWER-OF-2 VALUES OF $n$

When the number of GPUs is not a power of 2, StragglAR becomes more challenging to implement because GPUs can possess multiple active chunks at any point in time. To handle the even, non-power-of-2 case, we use a similar matching-based approach to construct the schedule in every round, but instead connect vertices $u$ and $v$ with weight 2 if $u$ possesses a chunk that $v$ needs and $v$ possesses a chunk that $u$ needs, and with weight 1 if only $u$ possesses a chunk that $v$ needs, but the opposite is not true (or vice versa). Then, we run well-known polynomial-time algorithms (Edmonds, 1965a;b) to compute the maximum weight matchings per round in order to generate the schedule. Thus, offline schedule generation when $n$ is not a power-of-2 has a higher runtime, but is still polytime computable.

For even, non-power-of-2 values of $n$, we do not prove a bound on the $\alpha-\beta$ cost. However, we still synthesize and evaluate these schedules in §4 and typically find schedules that complete in $\sim n+2\log n-2$ rounds in practice, which still results in lower $\beta$ cost than baselines. The realized communication complexity of the schedules generated by StragglAR for non-power-of-2 $n$ is shown in the ideal-case performance of Fig. 9, and still outperforms all baselines using the analytical network simulator. We note that StragglAR does not provide schedules for odd values of $n$, but these settings are less common in large-scale distributed ML; further, StragglAR's schedules for even $n$ are all computable in polytime due to the graph matching structure.

We compare StragglAR's ideal and worst-case performance to baselines as the cluster size scales, in simulation with the analytical network model. We report the results in Fig. 9, which shows a similar pattern as for powers-of-2. In ideal scenarios, when the REDUCESCATTER precondition can be fully overlapped with the straggler delay, StragglAR significantly outperforms all baselines. For worst-case scenarios with no straggler delay, StragglAR's performance is slightly worse than baselines at small cluster sizes, but performs equally well at large cluster sizes due to its $2\times$ lower asymptotic communication complexity.

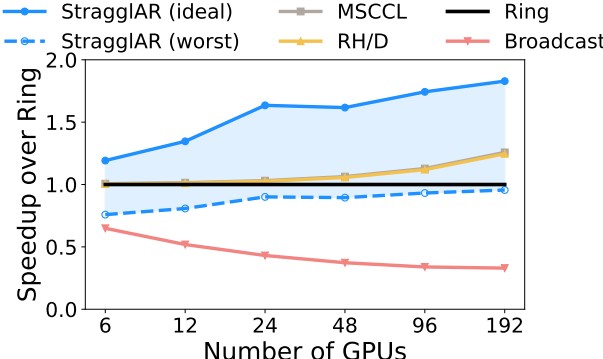

Figure 9: ALLREDUCE performance of different algorithms as cluster size $n$ scales, for even values of $n$ that are not powers of 2. The shaded region captures the range of StragglAR's possible performance, ranging from worst case (no straggler and hence no overlap) to ideal (full overlap with straggler).

# F  DISCUSSION OF BASELINES

We provide in-depth explanations and $\alpha-\beta$ costs of the baselines. Classical algorithms (*e.g.,* Ring, RH/D) achieve the known lower bounds for ALLREDUCE in networks with homogeneous bandwidth and thus remain the standard implementations in state-of-the-art CCLs (Hu et al., 2025). While newer works synthesize algorithms for irregular topologies and *heterogeneous networks* (Won et al., 2024; Wang et al., 2020; Zhao et al., 2024a), they do not provide new algorithms for homogeneous scale-up networks, aside from MSCCL (Cowan et al., 2023), which we compare to.

**Ring (Patarasuk and Yuan, 2009).** This is the standard bandwidth-efficient algorithm implemented by NCCL for ALLREDUCE. It divides the buffer into chunks of size $\frac{s}{n}$ and consists of $2(n-1)$ rounds in a ring-like communication pattern. The total runtime is $T_{Ring} = 2(n-1)\cdot\alpha + 2\frac{n-1}{n}s\cdot\beta$.

**Recursive halving/doubling [RHD] (Bruck et al., 1994).** Also called the *Butterfly* algorithm, this algorithm—like Ring—consists of two phases. In the first phase, the buffer is first divided in $1/2$ with one partner, then in $1/4$ with another partner, *etc.* to achieve a REDUCESCATTER. Then, a mirror-image ALLGATHER completes to propagate all fully reduced chunks. The total runtime is $T_{RHD} = 2 \log n \cdot \alpha + 2\frac{n-1}{n}s \cdot \beta$.

**Broadcast.** This is a *straggler-aware* baseline that assumes the non-stragglers complete an ALLREDUCE ($2\times$ the time as REDUCESCATTER) during the straggler delay. After the straggler is ready, it exchanges its *entire* buffer with any other rank to fully reduce the entire buffer and then initiates a broadcast with $\log n$ rounds and $s$ bytes per round. The total ideal-case runtime is $T_{Bcast} = \log n \cdot \alpha + \log n \cdot s \cdot \beta$ while the worst-case runtime is $(2n + \log n - 4) \cdot \alpha + (\log n + 2\frac{n-2}{n-1}) \cdot s \cdot \beta$.

**MSCCL (Cowan et al., 2023).** This is a new algorithm synthesized for switched scale-up domains, such as NVIDIA DGX. The algorithm, called *allpairs*, consists of just two rounds. The buffer is divided into $n$ chunks. In the first round, each GPU is responsible for reducing one chunk (*i.e.,* REDUCESCATTER), and sends all other chunks to other GPUs simultaneously, thereby splitting the link bandwidth among $n-1$ parallel connections. The next step is a mirror-image ALLGATHER, where all GPUs communicate simultaneously to receive all reduced chunks. Thus, the total runtime is $T_{MSCCL} = 2 \cdot \alpha + 2\frac{n-1}{n}s \cdot \beta$.

# G   EVALUATION ON PERLMUTTER WITH 4 GPUS

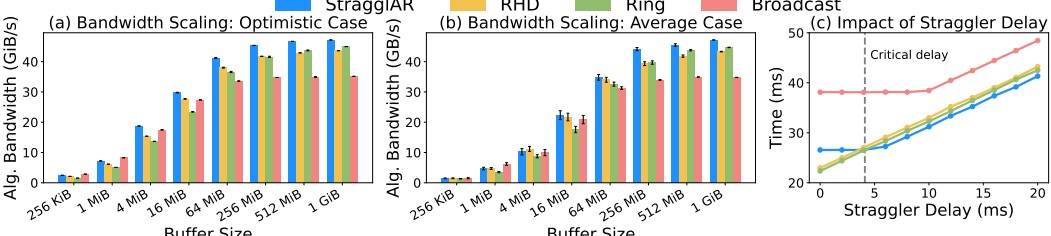

Figure 10: ALLREDUCE performance on a 4-GPU node in Perlmutter. (a) shows the optimistic use case, where it is assumed that the REDUCESCATTER can complete within the straggler delay time. (b) assumes a straggler delay of 9.45 ms, the average from our experiments. (c) fixes the buffer size at 1 GiB and varies the straggler delay.

We show the results from running on one node of the Perlmutter supercomputer (NERSC, 2025) in Fig. 10. While StragglAR still consistently outperforms baselines across buffer sizes in the 4-GPU setup, its gains are less substantial than for the 8-GPU setup. This is expected, and the results are in line with the theoretical results in §4 because StragglAR's performance advantage *improves* as $n$, the size of the GPU cluster, scales. (Intuitively, this is because the proportion of "work" that is offloaded to the precondition is higher as $n$ scales.) Fig. 10b indicates similar performance on the 4-GPU node even with both the average and masked delays, suggesting the REDUCESCATTER precondition can be effectively masked in typical scenarios.

# H   NCCL TUNING BEHAVIOR

As we discuss in §4, the performance of `ncclSend()`/`ncclRecv()` deviates from the $\alpha - \beta$ cost model's expected performance for some specific buffer sizes due to heuristics and protocol-switching behavior implemented internally by NCCL. We run the baseline `nccl-tests` (NVIDIA, 2024b), a benchmarking suite provided by NVIDIA for evaluating in-built NCCL performance, across different data sizes and plot the results in Fig. 11. The region from 64 MiB-256 MiB deviates from expected behavior under $\alpha - \beta$ costs because sending $2\times$, and even $4\times$, the amount of data results in almost no change in communication time with the P2P API. (Under the $\alpha - \beta$ model, the time should consistently scale linearly with the send/recv data size, instead of step-wise as shown in Fig. 11.) This is why the Direct and MSCCL ALLREDUCE baselines outperform other algorithms at 256 MiB, as the time to send an entire 256 MiB buffer is only marginally higher than sending smaller chunk sizes, which

bandwidth-efficient algorithms like Ring, RHD, and StragglAR do. We do not observe this finding to generalize to other architectures, and it may be a relic of the NCCL version, CUDA version, and specific environment configured by RunPod for `ncclSend()`/`ncclRecv()`. Also, outside of the 64-256 MiB range, the performance closely reflects the $\alpha-\beta$ model, and our findings in §4 reflect this. This anomalous behavior is also well documented by recent works, which even propose methods to optimize tuning parameters for NCCL (Hu et al., 2025; Wang et al., 2025).

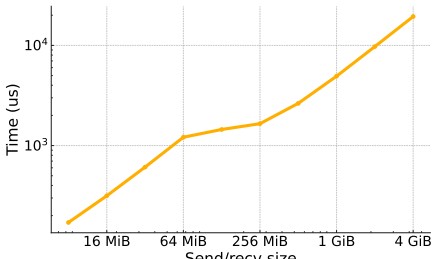

Figure 11: `nccl-tests` benchmarking of NCCL P2P API on RunPod server with 8 A100 GPUs.

## I END-TO-END TRAINING EXPERIMENTS ON MULTI-GPU SERVERS

We evaluate StragglAR using data-parallel (DP) fine-tuning of the Llama3.2-3B model in PyTorch on a DGX A100 VM (8-GPU setup) on RunPod. In this evaluation, we enable ALLREDUCE algorithms to run in the wild, calling implementations directly from PyTorch on a real hardware testbed. To do so, we compile our C++ and CUDA implementations (using the NCCL P2P API, exactly as described in §4) into a custom package, and enable `dist.all_reduce()` to call these implementations using PyTorch Distributed's C++ extension capabilities (PyTorch, 2022). We compare StragglAR to the Ring algorithm, the most well-known and widely used bandwidth-efficient baseline, under this scheme. Using standard PyTorch profiling tools, we identify device (GPU) 6 to be the most frequent straggler on our server, as device 6 was most frequently the straggler in the profiling logs. At the start of training, we pass the straggler rank to the CUDA runtime to ensure StragglAR executes correctly. Then, training simply proceeds as per usual with PyTorch.

Next, we fine-tune Llama3.2-3B using the same data-parallel setup on 8 GPUs with the global batch size of 32 for 100 iterations to determine the overall training efficiency gains with StragglAR. Fig. 12 shows that even these first 100 iterations, StragglAR reduces training time by 7 seconds. These gains suggest that training at longer time scales—for example, days—can shave off several hours of training. Further, we note that StragglAR's loss curve is just the Ring ALLREDUCE curve shifted to the left: the $y$-coordinates are exactly the same because both algorithms ensure *complete* data buffers are reduced, ensuring no change in model convergence. Instead, StragglAR enables an unmodified, but faster ALLREDUCE to the application layer, resulting in greater training efficiency and faster time-to-convergence. Fig. 12 captures the wall-clock time, *i.e.,* all compute and communication operations in training, so the results may improve in larger-scale settings (as shown in simulations in §J) or with even larger model sizes where communication becomes a greater bottleneck.

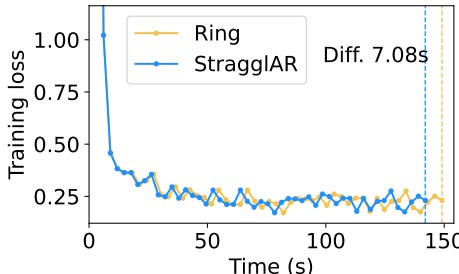

Figure 12: End-to-end evaluation for Llama-3.2-3B fine-tuning on a RunPod DGX A100.

## J  END-TO-END TRAINING SIMULATIONS AT SCALE

We use the FlexNet simulator used by prior work (Wang et al., 2023a)—an augmented version of FlexFlow (Jia et al., 2019; FlexFlow)—to determine 3D parallelism schemes for transformer training at even larger scales. Specifically, FlexNet outputs the task graph of compute and communication tasks per training iteration. We simulate training the BERT-large transformer model (Kenton and Toutanova, 2019) as we vary both the number of GPUs and the straggler delay. We provide end-to-end training iteration time reported by FlexNet and using the analytical network simulator to predict the communication time (for both collective and point-to-point communication) as an input. Fig. 13 shows end-to-end results assuming $\alpha=3\mu s$ (based on (Microway, 2024)) while Fig. 14 shows these results assuming $\alpha=0.7\mu s$ (based on (Shah et al., 2023)). In general, StragglAR outperforms baselines as the number of GPUs scales when there is even marginal delay of 0.5 ms or higher. When there is no delay, StragglAR suffers because *none* of the REDUCESCATTER precondition is overlapped with the straggler delay. In Fig. 13, StragglAR either performs more poorly than or evenly with RHD (depending on the straggler delay) for $n=64$ because the assumption of $\alpha$ is higher (so, algorithms like RHD that minimize the number of rounds are preferred when $n$ is large). In contrast, Fig. 14 assumes a lower value of $\alpha$ that is reported by prior published work (Shah et al., 2023), thereby enabling StragglAR's bandwidth efficiency to shine even at large values of $n$. The exact buffer sizes and compute and communication kernels are directly outputted by FlexNet, and we evaluated with batch sizes of 16, 32, 64, and 128 on 8, 16, 32, and 64 GPUs, respectively. These batch sizes are based on typical fine-tuning use cases that opt for smaller batch sizes on smaller datasets (Chung et al., 2024; Liu et al., 2019; Touvron et al., 2023).

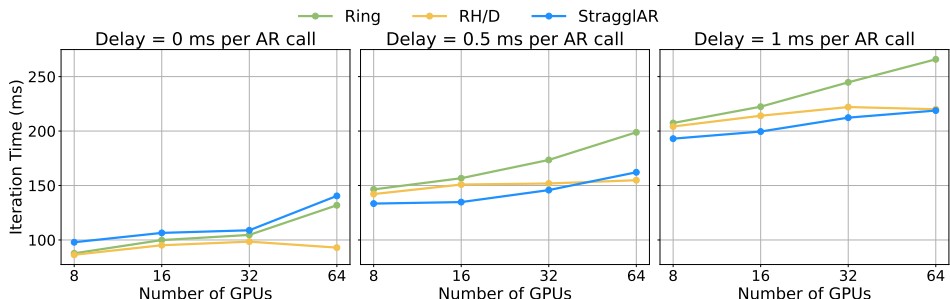

Figure 13: Runtime per BERT training iteration using the FlexNet simulator with $\alpha=3\mu s$.

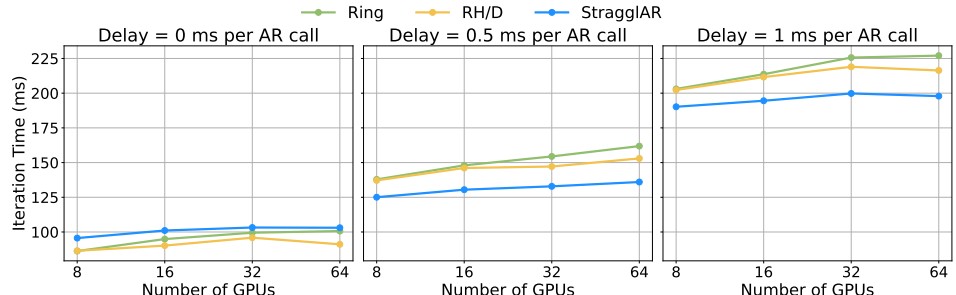

Figure 14: Runtime per BERT training iteration using the FlexNet simulator with $\alpha=0.7\mu s$.

