# OpenReview forum: "Efficient AllReduce with Stragglers"
_ICLR.cc/2026/Conference — Submitted to ICLR 2026_

### Official Review · Reviewer_ihtY · 2025-10-31

**Soundness:** 3
**Presentation:** 3
**Contribution:** 2
**Rating:** 4
**Confidence:** 4

**Summary:**

This paper introduces Str-glAR, a novel AllReduce algorithm designed to mitigate the performance bottleneck caused by stragglers in distributed machine learning. The core idea is to utilize the idle time incurred while n-1 nodes wait for the straggler by proactively executing a ReduceScatter operation as a pre-processing step. Once the straggler is ready, the system executes a specialized, lightweight second phase of communication to complete the final, mathematically exact global aggregation.

**Strengths:**

1. Stragglers are a long-standing and impactful bottleneck in large-scale distributed systems. This paper tackles the problem not by compromising on correctness (approximation) or synchronicity (asynchrony), but by fundamentally redesigning the communication algorithm itself while preserving both.

2. The most compelling contribution is the insight that by challenging the implicit assumption of a simultaneous start, it is possible to achieve a lower communication cost than the ~2Sβ bound.

3. The evaluation is robust and covers multiple layers of the system stack, from low-level communication performance (microbenchmarks) and real-world application impact (end-to-end LLM training) to scalability (simulations).

**Weaknesses:**

1. Failure to Clearly Position its Novelty with Insufficient Discussion of Related Work: The paper's most significant weakness is its failure to adequately differentiate its core idea from the decades-long body of research on "overlap" in systems. The high-level concept of "doing useful work while waiting" is a classic optimization principle. The authors must more rigorously distinguish their "intra-algorithm overlap" from traditional "compute-communication overlap" in the related work section.

2. Overly Idealized Discussion of System Trade-offs: The paper emphasizes its lossless nature but does not sufficiently address the trade-offs between its benefits and the complexity it introduces.

3. Compared to simply dropping a straggler (which is lossy but simple to implement), Str-glAR introduces a complex two-phase protocol. The paper should more honestly discuss the trade-off where a simpler, approximate method might be preferable in latency-critical applications that can tolerate some error.

4. The ReduceScatter phase itself has a non-trivial cost. If the straggler's delay is insufficient to fully hide this cost, the net benefit will diminish. The analysis of scenarios with short delays is not deep enough.

5. The algorithm's design and theoretical analysis are heavily predicated on a single, well-defined straggler. Real-world completion times may follow more complex distributions (e.g., heavy-tailed). The paper needs to discuss the algorithm's behavior and trigger mechanism when a distinct straggler does not exist.

**Questions:**

1. Could you explicitly contrast Str-glAR's "intra-algorithm overlap" with the classic "compute-communication overlap" in your introduction and related work? Please clarify why your contribution should be considered a fundamental algorithmic innovation rather than another scheduling heuristic.

2. The execution time of the ReduceScatter phase, T_rs, is a critical overhead. Can you provide a more quantitative analysis of how the relationship between the straggler delay T_delay and T_rs impacts the final performance? Is there a tipping point where, if T_delay < T_rs, Str-glAR could perform worse than a standard Ring-AllReduce due to protocol overhead?

3. Your end-to-end experiments rely on a pre-identified, persistent straggler. How would the system adapt if the straggler's identity changes frequently between iterations? What is the runtime overhead of dynamically detecting the first n-1 arrivals and dispatching the appropriate communication strategy?

4. How would Str-glAR perform in the presence of multiple stragglers (e.g., two nodes that are significantly slower than the other n-2)? Can the current protocol be naturally extended to handle this, or would it require a completely new algorithmic design?

---

> ### Author Response · Authors · 2025-11-19
> **Author Response [1/2]**
>
> Thank you for your feedback. We are encouraged that you find our contribution of surpassing the known $2s\beta$ lower bound for bulk-synchronous AllReduce by leveraging straggler delays to be compelling, and that our evaluation is robust. We address your concerns below and have updated the uploaded paper to reflect the changes.
>
> ## Comparison with classic compute-communication overlap
> We discuss and compare to related work in this area at the end of $\S A$ (see heading “Compute-communication overlap”). Our work develops on classic compute-communication overlap by taking this concept one level deeper: compute-communication overlap _within_ the collective communication algorithm. In summary, existing strategies for compute-communication overlap operate at the operator or block level. Thus, for communication operations, these techniques rely on the same set of collective algorithms implemented by the underlying collective communication library (CCL). In contrast, our work considers overlap within the communication operation and requires a fundamentally novel approach, as existing bandwidth-optimal collective algorithms rely on all GPUs being ready before beginning the collective operation. Because our technique begins the AllReduce _before_ the straggler reaches the synchronization barrier, it required designing an algorithm that could begin from an asymmetric precondition ($n-1$ GPUs that have completed a ReduceScatter vs. one that has not) while still outperforming bandwidth-optimal algorithms like Ring. Thus, our contribution is a novel parallel communication algorithm for AllReduce, as opposed to a scheduling heuristic or compiler optimization; we have updated $\S A$ to expand on this comparison.
>
> ## Quantitative analysis of ReduceScatter overhead and systems trade-offs
> In Figures 5(c) and 5(f), we provide the value of this “tipping point” straggler delay (i.e., the delay needed to outperform Ring), which we call the __critical delay__, for a buffer size of 4 GB. We also discuss this phenomenon in detail in $\S 4$ under the heading “Varying straggler delay,” where we stated that “the critical delay approaches zero as the size of the GPU cluster increases due to StragglAR’s efficient scaling.”
>
> We have added $\S B$ to provide a precise quantitative analysis of this relationship along with relevant results in Fig. 7. Below, we summarize our analysis in $\S B$ of the relationship between $T_{delay}$ and $T_{RS}$. While the ideal performance bound is achieved when $T_{delay} \geq T_{RS}$, StragglAR will often outperform baselines even when the straggler delay is not long enough to fully overlap the ReduceScatter. Specifically, StragglAR only underperforms the Ring algorithm if $T_{RS} + T_{SAR} > T_{delay} + T_{ring},$ where $T_{ring}$ is the Ring algorithm's complexity and $T_{SAR}$ is our schedule's complexity.
>
> In $\S B$, we solve this expression for $T_{delay}$ and prove that StragglAR outperforms baselines as long as $T_{delay} > (\log{n}-2)\alpha + \frac{log{n}}{n}s\beta$, which is substantially smaller than the ReduceScatter time of $T_{RS} = (n-2)\alpha + \frac{n-2}{n-1}\beta$.  Even with $n=256$ GPUs, $(\log{n}-2)\alpha$ results in negligible sub-millisecond cost (see $\S B$ for more details), so we focus on the bandwidth term of  $\frac{\log{n}}{n}s\beta$. The coefficient $\frac{\log{n}}{n}$ is a fraction that is always less than 1, decreasing in $n$, and approximately zero for large values of $n$. Thus, at large scales, StragglAR performs on par with the Ring algorithm, even without any straggler delay, and the straggler delay required for competitive performance is still very small (a small fraction of the time to simply send the buffer to a peer) even for smaller GPU clusters. For example, the critical delay for 1 GB on an 8-GPU DGX H100, with 450 GB/s P2P bandwidth, is 0.83 ms according to this expression. This means that _less than one millisecond of straggler delay_, well within the delay distribution we observe empirically, is required for StragglAR to outperform the Ring algorithm even in this smaller-scale setting. To address your concerns about operational complexity, we have added more discussion about this to the Limitations section in $\S 4$ of the paper. As described in $\S 4$, the two-phase synchronization is handled by our implementation and is not something the developer has to worry about.
>
> **The performance bounds we derive are not for a single, well-defined straggler. As we discuss in the next two sections, our performance bounds are general: they apply to dynamic stragglers and multiple stragglers, since all of these scenarios are parameterized by the straggler delay (i.e. the delay between the last and second-to-last ranks to reach the synchronization barrier).** Our complexity analysis is based on this delay and thus applies generally to both dynamic and multiple stragglers.

---

> > ### Author Response · Authors · 2025-11-19
> > **Author Response [2/2]**
> >
> > ## Dynamic stragglers
> > StragglAR is a fundamental parallel communication algorithm, and supports both dynamic and persistent stragglers. In our end-to-end evaluation, we allow the workload to run unmodified, and the straggler’s rank naturally varies over iterations. Even with dynamic stragglers, our approach does not require straggler detection: StragglAR can be executed as soon as $n{-}1$ GPUs reach the synchronization barrier. This only adds a conditional statement on the host to execute the appropriate schedule. If the final GPU arrives immediately after (i.e., no straggler delay), then StragglAR simply achieves its worst-case performance bounds, which match that of baselines at scale (see $\S 3.2$, Table 1, Figure 6(c), and $\S B$). If the final GPU arrives later (i.e., it is a straggler) StragglAR’s performance improves, outperforming baselines as long as the delay is longer than the critical delay, which is approximately 0 ms for large $n$ and still small even for lower values of $n$ ($\S B$). Our benchmarking experiments capture this performance range from all angles—across buffer sizes, straggler delays, and different hardware environments.
> >
> > ## Multiple stragglers
> > While GPU arrival times at the synchronization barrier form a distribution (Fig. 2(a)), the identity of the straggler is discrete: it is the GPU with the maximal delay. Therefore, by definition, multiple stragglers can only occur if multiple devices reach the synchronization barrier _simultaneously_. Otherwise, even if multiple GPUs are delayed, there will always be one GPU that is delayed longer compared to the rest, constituting the straggler. In Fig. 2(a), we plot the time difference, $\delta t$, between when the last and the second-to-last GPUs reach the synchronization barrier. Multiple simultaneous stragglers, or no straggler whatsoever, correspond to $\delta t =0$ (or similarly small values), as the second-slowest and slowest ranks would reach the synchronization barrier around the same time. Since GPU execution times are continuous variables, the likelihood of simultaneous stragglers is exceedingly low (effectively 0 probability). This is further confirmed by the 0 probability mass at a straggler delay ($\delta t$) of 0 ms in Fig. 2(a).
> >
> > Despite the low probability, dealing with multiple simultaneous stragglers remains a challenging open problem, as not only do stragglers have to exchange chunks with other ranks, but they also have to use some rounds to exchange chunks with each other. This means that not all rounds are productive in a one-dimensional topology (since chunks are not fully reduced until all stragglers have transmitted their chunk), unlike StragglAR. This poses a challenge to leverage the delay to outperform baselines and offers an exciting opportunity for future work.
> >
> > ## Dropping the straggler’s data
> > We discuss this scenario in $\S A$ and further address it with new text in the Limitations section in $\S 4$. Strategies that simply drop the straggler’s data crucially rely on the specific application’s error tolerance and are often only preferable for _severe stragglers_, where the delay is so significant that it outweighs the benefits of using the straggler’s data. For example, earlier works (~2011) introduced fully asynchronous data-parallel training [1], but these approaches have not been adopted widely nowadays, especially due to strict correctness requirements in tensor parallelism [2]. However, some applications may have stringent latency requirements that supersede concerns about lossy and erroneous techniques, and could benefit from simply dropping the straggler’s data.
> >
> > If we have sufficiently addressed your concerns, we would be grateful if you would consider increasing your rating. We are happy to answer any questions.
> >
> > [1] Feng Niu, et al. HOGWILD!: A Lock-Free Approach to Parallelizing Stochastic Gradient Descent. _NeurIPS, 2011_.
> >
> > [2] Deepak Narayanan, et al. Efficient large-scale language model training on GPU clusters using megatron-LM. _ACM Supercomputing Conference (SC), 2021_.

---

> > > ### Comment · Reviewer_ihtY · 2025-11-27
> > >
> > > 2.Multiple stragglers.
> > > I find the rebuttal regarding multiple stragglers---specifically the claim that "multiple stragglers can only occur if multiple devices reach the barrier simultaneously," which has zero probability---to be theoretically convenient but practically flawed.
> > > The critical issue is not whether two stragglers arrive at the exact same nanosecond, but rather the strict requirement that your algorithm needs $n-1$ ready nodes to trigger the pre-processing phase. In a realistic scenario (e.g., an 8-GPU cluster) where two GPUs are delayed due to shared resource contention or thermal throttling, the ``fast'' group consists of only 6 GPUs.
> > > Consequently, StragglAR cannot execute its precondition until the second slowest GPU arrives to meet the $n-1$ threshold. This means the system must still wait idly for the second straggler, rendering the algorithm ineffective for the duration of the first straggler's delay regardless of exact simultaneity. This strict dependency makes the approach brittle in real-world clusters where stragglers often appear in groups, fundamentally limiting its robustness compared to simpler straggler-dropping or asynchronous methods.

---

> > ### Comment · Reviewer_ihtY · 2025-11-27
> >
> > I thank the authors for their detailed response and the updated revisions. I have carefully reviewed the rebuttal, particularly regarding the positioning of novelty, the analysis of critical delay, and the handling of multiple stragglers.
> > However, after considering the authors' arguments, I maintain my original score. Here is my reasoning based on your rebuttal:
> > 1. Comparison with classic compute-communication overlap
> > While I appreciate the authors' attempt to distinguish "intra-algorithm overlap" from classic "compute-communication overlap," I remain unconvinced that this represents a fundamental algorithmic breakthrough. Conceptually, executing a ReduceScatter on available nodes while waiting for a straggler is a logical engineering optimization—essentially "eager execution" of the first phase of a Ring-AllReduce. The core issue is that this feels like applying a decades-old HPC optimization principle (doing useful work during idle time) to a modern context (LLM training). While the context (Scale-up/NVLink) is current, the algorithmic insight appears to be a specific heuristic optimization rather than a new paradigm solving the unique new problems brought by LLMs.

---

> ### Author Response · Authors · 2025-11-27
>
> > I remain unconvinced that this represents a fundamental algorithmic breakthrough.
>
> We respectfully disagree. We believe that designing a novel AllReduce algorithm that surpasses the known $2s\beta$ lower-bound for bandwidth-optimal AllReduce is a fundamental algorithmic breakthrough.
>
> > The core issue is that this feels like applying a decades-old HPC optimization principle (doing useful work during idle time) to a modern context (LLM training).
>
> Many prominent recent works use the decades-old principle of doing useful work during idle time to speed up LLM training [1,2,3,4,5]. In fact, this principle has enabled significant recent progress in systems for ML, such as minimizing pipeline bubbles in pipeline parallelism, identifying opportunities for compute-communication overlap, using one-sided primitives for expert parallelism, etc. Our work novelly extends this principle to a new domain: overlap within a collective algorithm. "Doing useful work during idle time" is a very broad principle, and there are many novel and specific ways to apply it to distributed ML that require careful reasoning and have resulted in significant progress in state-of-the-art LLM training systems. In particular, our approach leverages intra-algorithm compute-communication overlap while also competing with and surpassing the lower bound for AllReduce bandwidth complexity through polynomial-time collective algorithm design.
>
>
> [1] Overlap Communication with Dependent Computation via Decomposition in Large Deep Learning Models. _ASPLOS 2023_.
>
> [2] Breaking the Computation and Communication Abstraction Barrier in Distributed Machine Learning Workloads. _ASPLOS 2022_.
>
> [3] Centauri: Enabling Efficient Scheduling for Communication-Computation Overlap in Large Model Training via Communication Partitioning. _ASPLOS 2024_.
>
> [4] Lancet: Accelerating Mixture-of-Experts Training via Whole Graph Computation-Communication Overlapping. _MLSys 2024_.
>
> [5] COMET: Fine-grained Computation-communication Overlapping for Mixture-of-Experts. _MLSys 2025_.

---

> > ### Comment · Reviewer_ihtY · 2025-11-28
> >
> > I appreciate the authors' persistence and the citation of recent work. However, the latest response reinforces my concerns regarding the practical robustness of the proposed method.
> >
> > 1. The "Final GPU" vs. The "Optimization Gap" (The $N-1$ Dependency)}
> >
> > The authors argue that there is inherently a single "final" straggler due to continuous time distributions. This is mathematically trivial but structurally irrelevant to the system's performance gain.
> >
> > Crucially, StragglAR's performance gain is not determined by when the final straggler arrives, but by the time delta between the $N$-th GPU (slowest) and the $(N-1)$-th GPU (second slowest).
> >
> > Since the algorithm requires $N-1$ nodes to be ready to trigger the \texttt{ReduceScatter} precondition: 1) If the "final" straggler is delayed by $100$ ms, but the "second slowest" straggler is also delayed by $99$ ms (a very common scenario with shared resource contention, switch congestion, or node-level throttling), the available optimization window is only 1 ms. 2) In this scenario, StragglAR provides negligible benefit over the baseline (likely less than the protocol overhead $T_{overhead}$) while incurring complexity.
> >
> > The authors' insistence that "stragglers appearing in groups" is not a problem ignores this strict dependency on the gap, not just the absolute arrival time. The algorithm is brittle because it requires the straggler to be uniquely slow compared to all other peers by a significant margin (at least $T_{rs}$).

---

> ### Author Response · Authors · 2025-11-27
>
> > This strict dependency makes the approach brittle in real-world clusters where stragglers often appear in groups
>
> While we understand your concern about stragglers appearing in groups, our measurements and prior work [1] suggest a different picture. In Fig. 2(a), we observe a natural distribution of GPU execution times that inherently results in there being a "final" GPU to reach the synchronization barrier, and prior work that conducts large-scale ML training measurements highlights a similar phenomenon with a "final" straggler GPU [1]. However, even if stragglers appear in groups, our algorithm can either leverage the remaining delay to execute the ReduceScatter pre-condition or defaults to its worst-case performance bounds, which are still bandwidth-optimal. We acknowledge and note in the Limitations section that in cases with very severe stragglers, it might be preferable to drop the straggler's data, but this results in irreparable data loss and is infeasible for tensor-parallel AllReduce. However, our experiments do not reveal this phenomenon, and we believe our algorithm is useful in the vast majority of cases in which GPU arrival times to the synchronization barrier naturally vary, as further confirmed by our end-to-end training speedups.
>
> [1] Understanding Stragglers in Large Model Training Using What-if Analysis. _OSDI 2025_.

---

> > ### Comment · Reviewer_ihtY · 2025-11-28
> >
> > 2. "Algorithmic Breakthrough" vs. Engineering Optimization
> >
> > The classic $2s\beta$ lower bound is predicated on the assumption of a synchronous start. Surpassing this bound by relaxing the assumption (i.e., allowing $N-1$ nodes to start early while waiting) is a valid system optimization strategy (Eager Execution); however, describing it as a "fundamental algorithmic breakthrough" may be too strong. It is a heuristic application of overlap, tailored for a specific relaxed constraint, rather than a new fundamental algorithm that solves the problem under the original constraints.

---

> ### Author Response · Authors · 2025-11-28
>
> Thanks for engaging with us in the discussion.
>
> > The algorithm is brittle because it requires the straggler to be uniquely slow compared to all other peers by a significant margin (at least $T_{rs}$
>
> The algorithm outperforms the bandwidth-optimal lower bound as long as the straggler is delayed by the _critical delay_, which is **less than** $T_{rs}$, as we discuss in the rebuttal and prove in $\S B$. The critical delay is $\sim \frac{\log{n}}{n}s\beta$, which is effectively zero for large $n$ and less than 1 ms for $n \geq 8$ with a 1 GB buffer on the DGX H100 (Fig. 7(a)). Thus, StragglAR can provide speedups even in the scenario you described with a 1 ms straggler delay between the final two GPUs, and requires effectively no delay (i.e., critical delay $\approx$ 0) between the slowest and second-slowest GPU to enable speedups for large cluster sizes. In all of our experiments on multi-GPU clusters (across different domains, cloud VMs and supercomputer nodes), we do not see evidence for the described phenomenon of two GPUs being equally delayed (Fig. 2(a)). This has also not been documented as a common cause of stragglers in prior work that has studied straggler effects in large-scale ML training workloads [1].
>
> [1] Understanding Stragglers in Large Model Training Using What-if Analysis. _OSDI 2025_.

---

### Official Review · Reviewer_HXX7 · 2025-10-31

**Soundness:** 3
**Presentation:** 3
**Contribution:** 3
**Rating:** 6
**Confidence:** 4

**Summary:**

This paper proposes StragglAR, a new AllReduce (AR) algorithm that leverages straggler delay to preemptively schedule ReduceScatter (RS) among healthy ranks. For large data packets, it shows significant speedup (~25%) over Ring-AR without any precision loss. As a fundamental, lossless communication primitive, it has broad applicability for technologies like data and tensor parallelism.

**Strengths:**

1. Well-motivated: This paper works on a practical problem prevalent in distributed training with stragglers. It has strong motivation with empirical and literature support for straggler prevalence. It has an innovative low-level AR redesign.
2. Solid Work: Solid algorithm design with clear exposition. The experiments on a small-scale environment provide preliminary verification of the proposed method's effectiveness under varying straggler delays. The end-to-end speedup is remarkable.

**Weaknesses:**

1. Unfair Complexity Analysis: The complexity analysis is not general, as it only considers the ideal case where RS execution is hidden by straggler latency. This makes the comparison with other AR algorithms unfair.
2. Missing Baseline: The paper omits a comparison with a critical and more recent baseline, MSCCL++ (https://arxiv.org/pdf/2504.09014), making it difficult to assess its incremental contribution.
3. Figure 7 shows that the straggler delay CDF varies by environment, yet Figures 5(b) and 5(e) use the same average delay. Why not use environment-specific average delays, especially given the hardware differences (H100 vs. A100)?
4. The end-to-end experiments in Section 4.2 lack discussion on model choice, use models that are not large-scale, and don't sufficiently prove that the "Ideal Straggler Delay" is similarly flexible in other environments.

Minor:
5. Figure 12 appears to be missing from the paper.

**Questions:**

I would increase the rating if the authors addressed my concerns.

---

> ### Author Response · Authors · 2025-11-19
> **Author Response [1/2]**
>
> Thank you for your thoughtful feedback. We are encouraged that you find our algorithm design to be innovative, with strong empirical motivation and end-to-end speedups. We address your concerns below and have updated the uploaded paper to reflect the changes.
>
> ## Fair complexity analysis
> In the revised version, we have updated Table 1 to clearly state the complexity bounds for both best-case (straggler delay fully overlapped with RS) and worst-case (no overlap of RS) scenarios. As we discuss in $\S 3.2$, StragglAR’s worst-case complexity mirrors that of baselines at scale, while exhibiting $2\times$ speedups in the ideal case. Figure 6(c) visualizes the full range of performance, from worst to best case, and we have added a new section, $\S B$, to discuss how the critical delay required for StragglAR's speedup decreases, asymptotically approaching zero, as the GPU cluster size increases.
>
> ## MSCCL++ baseline
> We have updated the text in $\S 4$ and $\S A$ to discuss and cite MSCCL++, which provides a novel API for developers to write custom GPU communication operations. MSCCL++ is complementary to our work, as it is a library used to _implement_ different collective algorithms rather than synthesizing new algorithms (i.e., it is a competitor to NCCL and could be used to implement StragglAR or any other AllReduce algorithm). In fact, the AllReduce algorithm that MSCCL++ implements is precisely the MSCCL (ASPLOS, 2023) baseline that we compare to in our evaluation: Two-phase AllPairs, which is bandwidth-optimal on modern GPU scale-up networks. We hope that our added discussion of MSCCL++ addresses your concerns.
>
> ## Environment-specific straggler delays
> We appreciate your feedback about this. The original version used the average delay on A100s for both Figs. 5(b) and 5(e). We have fixed this in the new version and have updated Fig. 5(b) accordingly to use the average straggler delay on H100s. To obtain this delay, we first ran Llama-3.2-3B fine-tuning experiments over 10 independent runs on RunPod DGX H100 GPU servers and observed an average straggler delay of ~4.475 ms. We then repeated the benchmarking experiments for H100s, but instead used 4.475 ms as the average straggler delay. We plot the results in the updated Fig. 5(b), which exhibits a similar trend to that for A100s, with StragglAR exhibiting superior performance until 4 GB, where its performance reduces to match baselines. This is expected because the profiled straggler delay is slightly less than the 5.5 ms critical delay for a 4 GB buffer (Fig. 5(c)). However, we note that the straggler delay depends heavily on the hardware environment, software stack, and workload. Thus, the newly added quantitative analysis in $\S 3.2$ and $\S B$ provides a rigorous and general framework to understand the performance gains given any specific hardware environment.

---

> ### Author Response · Authors · 2025-11-19
> **Author Response [2/2]**
>
> ## Questions regarding the end-to-end evaluations
> **Model choice:**
> We chose three popular, recent, and open-source 3-4B parameter language models for the end-to-end evaluations. These were the largest models that could fit in GPU memory across our hardware environments for training while maintaining a reasonable batch size. We selected these models because they have many downloads in the past month on Hugging Face (444K for Llama-3.2-3B, 1.3M for Phi-3-mini, and 189K for Qwen-2.5-3B) and provide diversity across open-source language model providers (e.g., Meta, Microsoft, Alibaba) so that our results are generalizable. We have added a discussion of model choice to the text in $\S 4.2$. As noted by Reviewer GAq1 and demonstrated by extensive prior work [1, 2], the straggler problem becomes more severe at larger scales, suggesting that our results represent a conservative estimate of potential benefits.
>
> **Ideal straggler delay:**
> To address your concern about the flexibility of the ideal straggler delay across environments, we run new experiments to characterize this delay over different buffer sizes on a DGX A100 8-GPU server and plot the results in the newly added Fig. 6(b). The ideal straggler delay is simply the time to complete a ReduceScatter among $n-1$ GPUs. Therefore, it is dependent on the buffer size and the inter-GPU link bandwidth. Smaller buffer sizes and higher inter-GPU bandwidth reduce the ideal straggler delay. However, in our experiments with Llama-3.2-1B vs. Llama-3.2-3B, we also find that straggler delays are more significant with the 3B parameter model (which inherently requires a larger gradient buffer size). Due to natural empirical variation in straggler delays across models, datasets, training/inference configurations, and hardware environments, we provide a general analytical formulation of the ideal straggler delay to enable users to directly reason about the performance benefits of StragglAR for their workload. In the newly added $\S B$, we analyze critical delay, which is the straggler delay needed for StragglAR to outperform bandwidth-optimal baselines; the critical delay is even smaller than the ideal straggler delay. $\S B$ shows that the critical delay is $(\log{n}-2)\alpha +  \frac{\log{n}}{n} s\beta$. Since $\alpha$ cost is insignificant for large buffers (e.g., $(\log{n} - 2)\alpha$ is several  _microseconds_ for $n=256$, as shown in $\S B$), we focus on the $\beta$ cost. The coefficient on $s\beta$ is $\frac{\log{n}}{n}$, which is always a fraction less than 1, monotonically decreasing in $n$, and its value is approximately zero for larger values of $n$ (see $\S B$).
>
> Thank you for pointing out the missing figure in the Appendix. This was a LaTeX typo that made the figure numbering skip over 11. We have corrected this accordingly (i.e., there is no missing figure).
>
> If we have sufficiently addressed your concerns, we would be grateful if you would consider increasing your rating. We are happy to answer any additional questions.
>
>
> [1] Jinkun Lin, et al. Understanding Stragglers in Large Model Training Using What-if Analysis. _OSDI 2025_.
>
> [2] Ziheng Jiang, et al. MegaScale: Scaling Large Language Model Training to More Than 10,000 GPUs. _NSDI 2024_.

---

### Official Review · Reviewer_GAq1 · 2025-11-01

**Soundness:** 4
**Presentation:** 3
**Contribution:** 4
**Rating:** 8
**Confidence:** 4

**Summary:**

This paper presents an ALLReduce algorithm when stragglers exist.
It's found that multiple stragglers are found in real scenarios, which slows down the entire training.
StragglAR is an algorithm to make the non-straggler GPUs to make some forward progress, such that the time can be saved even though there exists a straggler.
Experiments are conducted with 1B and 3B model with up to 8 GPUs to show practicality.

**Strengths:**

**[S1]** Stragglers are causing real problems in datacenters, and this work could save a lot of monetary cost, which the paper directly presents.

**[S2]** This paper presents a solid algorithm with adequate proofs. I don't see any problems in the algorithm itself.

**[S3]** Quantitative bandwidth analysis, telling the readers how much benefit can be gained in arbitrary environments.

**Weaknesses:**

**[W1]** Comparison with potential existing algorithms would be beneficial. For example, the authors mention that there exists a tree-based algorithm (AdapCC). I did not check AdapCC, but a layman knowing a tree-based AR algorithm could design a AR/RS a tree such that the straggler participates as late as possible. I believe it can be a simpler alternative. Would it be possible to devise a performance model of this to compare?

**[W2]** The experiments are weak in scale and settings.
- The experiments are done up to 3B model, only over 8 GPUs. This is way too small compared to what people are using for training, which usually spans multiple nodes. This is both good and bad. Good because the communication time is usually longer and straggling is more likely to exist in larger settings, but at the same time, but bad because the straggler effect could be dwarfed by other communication overheads. This is related to the next subitem.

- The authors report a few tens of milliseconds of straggler delays. However, it's difficult to find what portion of it occupies within the entire AR, or the entire (single) training step. Such information will be important to motivate this work.

- The cause and characterization of the straggler could be further investigated. I guess it might be out of scope to find the cause of straggers, but as a reader, I am very curious about at least how the stragger delay would change when using various larger models, or more servers. If the delay is coming of the network or network interface, it could remain relatively independent of the model size, where the slowdown of the GPU itself could cause the straggler delay to scale larger. This is related to the author's method of simulating the delays.


**[W3]** Minor suggestion. I was confused multiple times by the use of the word 'buffer' to indicate the volume of the communication.
It sometimes seemed like the buffers are the overhead caused by the stragglAR algorithm. The buffer size indeed decides how much data is handled in a single CC transaction, but I don't think it's a common term for researchers in the field.

**Questions:**

Please see weaknesses.

---

> ### Author Response · Authors · 2025-11-19
> **Author Response [1/2]**
>
> Thank you for your thoughtful feedback. We are encouraged that you appreciate our technical contribution and believe that it addresses a real and impactful problem. We address your concerns below and have updated the uploaded paper with relevant details.
>
> ## Performance model for straggler-aware Tree algorithms
> Thank you for this suggestion. We analyze the performance of a “straggler-aware” variant of the pipelined Tree algorithm implemented by NCCL, which improves bandwidth efficiency over standard Tree algorithms [1].
>
> In the Pipelined Tree algorithm, each GPU splits its entire buffer $s$ into $k$ chunks and transmits these chunks in a pipelined manner. Note that for this algorithm to be straggler-aware, it must be a single binary tree with the straggler placed at the root. The double binary tree algorithm [1] that implements two trees—Tree 1, an arbitrary ordering over the GPU ranks, and Tree 2, which inverts intermediate nodes of Tree 1 as leaves of Tree 2—inherently requires every GPU to participate from the very beginning (since every rank is a leaf node in one of the two trees) and thus cannot be made straggler-aware.
>
> The Pipelined Tree algorithm’s performance is $(k + 2\log{n} )\alpha + 2s\beta + 4\log{n}(\frac{s}{k})\beta$, which exhibits higher bandwidth complexity than both Ring and Recursive Halving/Doubling by an additional $4\log{n}(\frac{s}{k})\beta$. (This is why libraries like NCCL implement Ring over the Pipelined Tree algorithm for large buffers [1].) The runtime expression for the Pipelined Tree AllReduce is derived from the following principles. (1) There are $k + 2\log{n}$ rounds total, since $2\log{n}$ additional rounds are required for the reduced chunk to traverse up (Reduce) and back down (Broadcast) the tree after the final chunk is sent in the $k$-th round. (2) In each round, $\frac{s}{k}$ bytes are sent, but due to the binary tree structure, the link bandwidth is split in half in each round as a result of simultaneous communication with both children in the tree. Thus, each data transfer in a round incurs $2\beta$ bandwidth cost per byte.
>
> Now, we consider overlapping the straggler delay with the initial part of the reduction. At best, the straggler delay can overlap with the initial rounds reducing up to the root, consisting of $(\log{n})\alpha + \log{n}(\frac{s}{k})(2\beta)$ cost. Subtracting this from the total cost of the Tree algorithm (since after overlapping, this part is no longer exposed communication), we obtain the _best-case_ performance of the straggler-aware pipelined Tree: $(k + \log{n})\alpha + 2s\beta + 2\log{n}(\frac{s}{k})\beta$. Even with the ideal straggler-delay overlap, this algorithm has slightly higher (by $2\log{n}(\frac{s}{k})\beta$) bandwidth complexity than the Ring and Recursive Halving/Doubling baselines that we compare to in the paper, and more than $2\times$ higher bandwidth complexity than StragglAR. We hope this provides sufficient evidence that even pipelined Tree algorithms are insufficient alternatives for bandwidth-efficient AllReduce with stragglers. [1] also provides numerous experimental results confirming that the Ring algorithm consistently outperforms the Tree algorithm for large buffer sizes in our targeted setting.

---

> > ### Author Response · Authors · 2025-11-19
> > **Author Response [2/2]**
> >
> > ## Experiment scale
> > We chose three popular, recent, and open-source 3-4B parameter language models for the end-to-end evaluations. These were the largest models that could fit in GPU memory across our hardware environments for training while maintaining a reasonable batch size. We selected these models because they have many downloads in the past month on Hugging Face (444K for Llama-3.2-3B, 1.3M for Phi-3-mini, and 189K for Qwen-2.5-3B) and provide diversity across open-source language model providers (e.g., Meta, Microsoft, Alibaba) so that our results are generalizable. We have added a discussion of model choice to the text in $\S 4.2$. Like prior work [2,3], we use the extensively validated and standard $\alpha{-}\beta$ simulator for analyzing performance beyond 8 GPUs. Prior work has also shown that straggler delays are even more significant for larger training jobs and models and present a critical bottleneck in scaling ML systems [4].
> >
> > We agree that there may be cases in which communication dominates straggler delays (e.g., if link bandwidth is very low), and have added discussion of these complexities in the Limitations section (end of $\S 4$) of the revised paper. However, with inter-GPU interconnect bandwidth continuing to grow rapidly [5], our experiments show that synchronization is an increasing and fundamental bottleneck for inter-GPU communication. Because StragglAR is bandwidth-optimal, it is well-suited for conditions with lower link bandwidth, where the per-byte data transfer cost is high. In fact, StragglAR performs on par with baselines at scale even if there is no or minimal straggler delay, as shown by the algorithm’s performance range in Fig. 6(c), the worst-case performance analysis in $\S 3.2$, and the newly added analysis regarding the critical delay in $\S B$.
> >
> > **End-to-end impact of stragglers:**
> > We appreciate your concern regarding the end-to-end impact of stragglers. We repeat the motivation experiments with multiple different models and VMs, and replicate our findings. We find that up to 23% of the second-to-last rank’s AllReduce time is spent waiting for the straggler, with some ranks spending up to 64% of their AllReduce time idling; we add these statistics to $\S 1$ of the main text. However, the impact of the straggler on end-to-end iteration time depends on the proportion of training time spent on AllReduce, which we find to be ~20% in our end-to-end experiments, but can vary widely depending on the model, software stack, protocol selection, and hardware environment. The end-to-end speedups reported in Table 2 of $\S 4.2$ further confirm that StragglAR directly reduces end-to-end training iteration time.
> >
> > **Causes of stragglers:**
> > Recent work has attempted to identify the root causes of stragglers through large-scale measurement studies [4]. The same prior work has shown that “most slowdowns are caused by computation operations, as opposed to communication” [4], which is corroborated by our own data that shows straggler delays resulting from slowdowns in preceding compute kernels on one of the GPUs. We believe that comprehensively understanding the root causes of stragglers remains a challenging open problem for future work. Since stragglers have been a consistent challenge over decades of research in distributed and parallel computing, we believe they are here to stay and require new techniques, such as StragglAR, to enable efficient communication in the presence of expected performance variability.
> >
> > ## Buffer size clarification
> > Thank you for this feedback. In the revised paper, we have clarified (at the beginning of $\S 3$) that we use the term “buffer” to denote the communication volume.
> >
> > We hope that we have sufficiently addressed your concerns, and we are happy to answer any questions.
> >
> > [1] Zhiyi Hu, et al. Demystifying NCCL: An In-depth Analysis of GPU Communication Protocols and Algorithms. _IEEE Hot Interconnects 2025_.
> >
> > [2] William Won, et al. ASTRA-sim2.0: Modeling Hierarchical Networks and Disaggregated Systems for Large-model Training at Scale. _ISPASS 2023_.
> >
> > [3] Xizheng Wang, et al. SimAI: Unifying Architecture Design and Performance Tuning for Large-Scale Large Language Model Training with Scalability and Precision. _NSDI 2025_.
> >
> > [4] Jinkun Lin, et al. Understanding Stragglers in Large Model Training Using What-if Analysis. _OSDI 2025_.
> >
> > [5] Adithya Gangidi, et al. RDMA over Ethernet for Distributed AI Training at Meta Scale. _SIGCOMM 2024_.

---

### Official Review · Reviewer_8vuX · 2025-11-04

**Soundness:** 2
**Presentation:** 3
**Contribution:** 2
**Rating:** 2
**Confidence:** 4

**Summary:**

Paper proposes a straggler-aware allreduce algorithm for synchronizing model weights among GPUs in distributed training. This is an important problem for large scale training as communication can be a significant bottleneck at scale and one slow GPU can slow down the whole training step. Overall my main concern is that the evaluation of the algorithm is weak, it’s not in realistic large scale setups and the benefits of the algorithm would not be significant in realistic environments. Finally, this is a distributes systems paper and ICLR may not be the best fit.

**Strengths:**

- Paper is about an important problem in enabling large-scale distributed training
- In the limited scenarios the algorithm is evaluated, it lowers the upper bound of  allreduce computation

**Weaknesses:**

- It is a distributed systems focused paper so an ML conference might not be a good fit in terms of its topic and audience. MLSys, ASPLOS or HPC conferences would be a better fit for this topic.
- The algorithm, upper bound calculations assume there is only one straggler. But in real scenarios the delays exhibit a distribution, rather than a binary choice of straggler vs non-straggler categorization (as also illustrated in Figure 2 in the paper). Furthermore, artificial straggler delays are used in evaluations in the end-to-end experiments, the GPUs are profiled ahead of time and a slow GPU is selected manually to represent as a straggler rank. This is not a realistic evaluation scenario representing delays at large scale (1000s of nodes).
- The evaluation setup assumes a uniform GPU-GPU bandwidth, it does not take into account topology of the network.
- The benefits of the algorithm increases when the straggler delay is significant ( it needs to be at least 6-8ms to give speedup). And the possible max delay in Perlmutter supercomputer is 8ms according to Figure 8. So StragglAR would not provide any speedup in a more realistic training setup.

**Questions:**

- How’s the algorithm bounds change when there are more than multiple stragglers?
- How would you change the algorithm to make it aware of the topology of the network?

---

> ### Author Response · Authors · 2025-11-19
> **Author Response [1/2]**
>
> Thank you for your feedback. We believe that ICLR is a good fit for our work, as it will encourage adoption of our ideas by a wide-range of distributed ML applications that rely on efficient inter-GPU communication. Our work falls under the “infrastructure, software libraries, hardware, etc.” topic in the ICLR 2026 Call for Papers. To help a fair assessment, we provide clarifications below on straggler effects, our evaluation setup, and standard assumptions of AllReduce algorithms; we have updated the uploaded paper to clarify some of these details. We also address areas where our evaluation and contributions have been misinterpreted.
>
> ## How multiple stragglers impact the performance of our algorithm
>  While GPU arrival times at the synchronization barrier form a distribution (Fig. 2(a)), the identity of the straggler is discrete: it is the GPU with the maximal delay. Therefore, by definition, multiple stragglers can only occur if multiple devices reach the synchronization barrier _simultaneously_. Otherwise, even if multiple devices are delayed, there will always be one device that is delayed longer compared to the rest, constituting the straggler. In Fig. 2(a), we plot the time difference, $\delta t$, between when the last and the second-to-last GPUs reach the synchronization barrier. Multiple simultaneous stragglers, or no straggler whatsoever, correspond to $\delta t =0$ (or similarly small values), as the second-slowest and slowest ranks would reach the synchronization barrier around the same time. Since GPU execution times are continuous variables, the likelihood of simultaneous stragglers is exceedingly low (effectively 0 probability). This is further confirmed by the 0 probability mass at a straggler delay ($\delta t$) of 0 ms in Fig. 2(a).
>
> In $\S 3.2$, we discuss how the algorithmic bounds change between best and worst-case scenarios, where the worst case involves no straggler delay (such as multiple exactly simultaneous stragglers). We further capture the entire performance range of StragglAR in Figs. 2(b) and 6(c), along with sensitivity analyses in Figs. 5, 6(a), and 6(b). StragglAR’s bandwidth complexity is $\sim s \beta$ in the ideal case where the ReduceScatter precondition is fully overlapped with the straggler delay, and $\sim 2s\beta$ in the worst case with no straggler delay. StragglAR’s worst-case complexity achieves the known bandwidth-optimal lower bound of $2s\beta$ for AllReduce while its ideal-case complexity surpasses this bound by $2\times$; we believe this is a fundamental and significant advancement. StragglAR achieves faster AllReduce in many settings, as _any non-zero straggler delay_ (which is expected, since it is statistically unlikely that two GPUs will finish exactly simultaneously) results in StragglAR outperforming its worst-case bound. To our knowledge, our work is the first to demonstrate that the decades-old $2s\beta$ AllReduce lower bound can be surpassed by introducing compute-communication overlap _within the collective algorithm_.
>
> **We would like to address a key misunderstanding: our end-to-end experiments do not use artificial straggler delays. Instead, we directly run the ML workload on the GPU cluster, and any straggler delays are those that are observed organically on the hardware platform.**

---

> > ### Comment · Reviewer_8vuX · 2025-11-27
> >
> > Thank you for the clarification and update about the worst case performance. For the end-to-end experiments, I understand that you do not inject delays. However you profile the VM's and identify the ones with persistent stragglers to run the algorithm. Is that correct? If so, this represent the speedups in the worst case scenario and Table-2 should state that.

---

> ### Author Response · Authors · 2025-11-19
> **Author Response [2/2]**
>
> ## Network topology
> Modern GPU scale-up networks provide homogeneous inter-GPU bandwidth (e.g., NVLinks, NVSwitch). CCLs such as NCCL therefore implement algorithms that are explicitly designed for homogeneous bandwidth (e.g., Ring), and these are the algorithms practitioners routinely use in this setting due to their bandwidth optimality. Our work adopts the same assumptions and targets this standard scale-up / rail-optimized scale-out regime in modern distributed ML.
>
> In contrast, there are some cases with heterogeneous link bandwidth, such as in multi-tenanted scale-out domains and wide-area deployments. These settings make use of an entirely different set of algorithms that are synthesized with purpose-built solvers using the specific network topology as input. In fact, the classic Ring algorithm is not suitable for heterogeneous network topologies, yet it remains the standard bandwidth-optimal AllReduce algorithm used today because contemporary scale-up domains and rail-optimized scale-out domains exhibit homogeneous GPU-GPU bandwidth; we target exactly the same setting. Extending our approach to heterogeneous interconnects is an interesting direction for future work but is outside the scope of this paper.
>
> **New experiments for scale-out domains:**
> Nonetheless, to highlight StragglAR’s applicability even to heterogeneous networks (e.g., multi-tenanted scale-out domains), we conduct a new experiment using 8 nodes from the Perlmutter supercomputer to compare StragglAR to the Ring algorithm. In a typical 3D-parallel sharding strategy, all GPUs of the same device ID across nodes would participate in an AllReduce. Thus, we measure the inter-node AllReduce time for device ID 0 for different buffer sizes with masked straggler delay. We report the mean speedups (10 iterations per buffer size) relative to Ring in the table below.
>
> |  **Buffer size**             |  64 MiB  |  256 MiB  |  1 GiB   |  2 GiB   |  4 GiB   |
> |------------------|----------|-----------|----------|----------|----------|
> | **Speedup (%)** |  35.5   |  36.1    |  35.7   |  35.7  |  35.7  |
>
>
> ## Straggler delay required for performance benefits
> The Perlmutter experiments in Figure 2(a) naturally feature a lower straggler delay due to the smaller model size, smaller cluster size (4 GPUs), and its status as a specialized supercomputer (as opposed to the cloud). The 5.5 and 7.5 ms critical delays are _specifically for a 4 GB buffer on the RunPod H100 and A100 DGX servers_. However, the critical delay depends on the hardware environment and the buffer size, so it is inaccurate to extrapolate performance by using the critical delay for the largest, stress-tested buffer size of 4 GB in one hardware environment to draw conclusions about another environment with a different buffer size. Nonetheless, straggler delays are lower in the experiments on the smaller Perlmutter nodes, so in this specific smaller-scale setting, our algorithm will typically only provide gains for smaller buffer sizes ($\leq$ 1 GiB) since they have lower critical delays. The straggler delay does not determine _whether_ our algorithm provides speedups, but rather, _for which buffer sizes_ it does so. As we discuss in $\S 3.2$, show in Fig. 6(c), and discuss in detail with additional experiments in the newly added $\S B$, StragglAR performs on par with baselines at scale _even when there is no straggler delay_.
>
> Our end-to-end experiments on H100 DGX servers involve organic straggler delays across different models and show that StragglAR provides speedups in realistic fine-tuning environments. While we cannot evaluate on “1000s of nodes,” in $\S 4.3$ we simulate performance at scale using the standard analytical model that is widely adopted in recent works for scaling analyses [1,2].
>
> If we have sufficiently addressed your concerns, we would be grateful if you would consider increasing your rating. We are happy to answer any questions.
>
> [1] William Won, et al. ASTRA-sim2.0: Modeling Hierarchical Networks and Disaggregated Systems for Large-model Training at Scale. _ISPASS 2023_.
>
> [2] Xizheng Wang, et al. SimAI: Unifying Architecture Design and Performance Tuning for Large-Scale Large Language Model Training with Scalability and Precision. _NSDI 2025_.

---

> > ### Comment · Reviewer_8vuX · 2025-11-27
> >
> > Regarding the delays for different buffer sizes, you mention "The 5.5 and 7.5 ms critical delays are specifically for a 4 GB buffer" and it is inaccurate to draw conclusions for different buffer sizes. Yet Figure 6 shows a straight line for "Avg. Delay in Experiments" and average delay is the same for different buffer sizes.
> >
> > Furthermore, smallest llm models have billions of parameters. A medium size model (i.e. 7B) would require 14GB of size for storing gradients in fp16/bf16 and so with data parallelism the allreduce size would be much larger than what's being experimented in the paper (i.e. Figure 5 / 6)

---

> ### Author Response · Authors · 2025-11-27
>
> Thank you for your feedback and engagement with our paper. We provide further clarifications and address your concerns below.
>
> ## Updating Table 2 to clarify the end-to-end speedups
> Yes, we do not inject delays in the end-to-end experiments (i.e., all straggler delays arise organically). On each VM, we first profile the workload to identify the GPU rank that is most often the straggler, and then pass this to the backend (see $\S 4.2$). In this sense, the end-to-end speedups can be interpreted as the worst-case speedups for the given hardware environment and workload, as additional speedups would be possible with dynamic detection of stragglers. **We have updated the caption of Table 2 to incorporate your feedback in the revised paper.**
>
>
> ## Clarifying the distinction between straggler delay and critical delay
> Figure 6 shows a straight line for the average delay because this is the average delay obtained from our earlier experiments running ML workloads on the specific hardware platform. The average straggler delay is not related to the buffer size, as it depends on a variety of factors that would cause a GPU to be delayed before the synchronization barrier: hardware variability, software overheads, thermal throttling, etc. In contrast, the _critical delay_ (e.g., 5.5 ms, 7.5 ms) depends on the buffer size because it is the minimum delay required for our algorithm to outperform the bandwidth-optimal lower bound (thus, it depends on the $\alpha-\beta$ costs and the buffer size, as shown by the formulation in $\S B$). As we discuss and prove in $\S 3.2$, $\S 4$, and $\S B$, the critical delay approaches zero as the size of the GPU cluster increases. We hope this clarifies the difference between the critical delay, which is a quantitative threshold value for our algorithm that depends on the buffer size, and the straggler delay, which is a measured value that arises organically from the GPU hardware/VM and does not depend on communication parameters such as the buffer size.
>
>
> ## Buffer sizes for benchmarking AllReduce
> While a 7B-parameter model would require 14 GB of memory for storing gradients in 16-bit precision, data-parallel AllReduce is typically executed with gradient bucketing. Gradient bucketing splits a large buffer into smaller buckets; the bucket size, rather than the entire model’s size, is the buffer size for each collective operation. (The entire model is not stored as a single tensor.) Thus, the range of buffer sizes (from 1 MB to 4 GB) we evaluate in our benchmarking experiments constitutes the standard range for benchmarking AllReduce performance, as used in both prior work [1,2] and official NCCL documentation [3].
>
> We hope we have addressed your concerns and are happy to answer any questions.
>
> [1] Meghan Cowan, et al. MSCCLang: Microsoft Collective Communication Language. _ASPLOS 2023_.
>
> [2] Guanbin Xu, et al. AutoCCL: Automated Collective Communication Tuning for Accelerating Distributed and Parallel DNN Training. _NSDI 2025_.
>
> [3] Ben Williams, et al. Understanding NCCL Tuning to Accelerate GPU-to-GPU Communication. _NVIDIA Technical Blog_. July 22, 2025.

---

### Author Response · Authors · 2025-12-04
**Discussion Summary**

## Summary
We propose StragglAR: a parallel algorithm for AllReduce that accelerates distributed training and inference by exploiting natural variation in GPU execution times (i.e., synchronization delays). StragglAR speeds up AllReduce by up to $2\times$ when synchronization delays occur (as we commonly observe) while maintaining performance competitive with bandwidth-optimal algorithms even when there is no delay.

**Synchronization is a real overhead**
* In our distributed training experiments, we observe significant synchronization delays, with some GPUs spending up to 64% of their AllReduce time idling while waiting for the slowest rank (i.e., the straggler).

* Bandwidth-optimal AllReduce algorithms (e.g., Ring) require all GPUs to be ready before the algorithm can begin and thus incur significant GPU idle time.

**Provably more efficient AllReduce with stragglers**
* StragglAR leverages the synchronization delay to surpass the $2s\beta$ lower bound for bandwidth-optimal AllReduce by $2\times$, providing fundamental speedups that _improve_ as cluster size scales.

* Even without any straggler delay, StragglAR retains competitive $\sim 2s\beta$ performance as baselines, as shown by the worst-case bound in $\S 3$.

**Evaluation on multi-GPU clusters, with end-to-end training speedups**
* We benchmark AllReduce performance across different multi-GPU environments (Perlmutter supercomputer, A100 DGX, H100 DGX).
* We conduct end-to-end training experiments across a variety of modern, widely used LLMs.
* We use the popular $\alpha$-$\beta$ simulator to evaluate performance on hundreds of GPUs, verifying that benefits grow with cluster size.

**A new paradigm for collective algorithm design**
* Our work introduces _temporal asymmetry_ to collective algorithm design, motivating future work in designing faster collective algorithms that relax bulk-synchronous assumptions.

## Reviewer concerns
**Reviewer HXX7's key requests**

Reviewer HXX7 provided key asks to increase their rating from a 6. We ran new experiments and edited the paper to address these.
* We include the worst-case complexity analysis in Table 1, highlighting that StragglAR provides significant upside when there is a synchronization delay ($2\times$ speedups) with minimal downside when there is not (performance on par with baselines).
* We ran experiments to characterize straggler delays on DGX H100s and benchmarked StragglAR using this average measured delay. We have updated Fig. 5 with these new results, which show patterns consistent with Fig. 6.
* We now cite and discuss MSCCL++ [Shah et al., _arXiv 2025_], which provides a novel API to implement communication primitives rather than synthesizing new collective algorithms. Like NCCL, it can be used to implement any of the algorithms discussed in the paper, including StragglAR. In fact, MSCCL++ implements the MSCCL AllPairs algorithm that we compare to in $\S 4$.

**Multiple stragglers [Reviewers ihtY, 8vuX]**

StragglAR’s performance bounds are parameterized by the straggler delay, i.e., the delay between the slowest and second-slowest GPU to reach the barrier. With no delay (e.g., multiple simultaneous stragglers), StragglAR achieves its worst-case bound, which is competitive with baselines. With even a small time difference between the arrival times of the slowest two GPUs, StragglAR outperforms its worst-case bound by overlapping part of the communication with the straggler delay.

* Our measurements in Fig. 2(a), the end-to-end experiments with organic straggler delays, and recent work about stragglers in large ML workloads [Lin et al., _OSDI 2025_] all demonstrate non-zero gaps between the slowest and second-slowest GPUs.

* For example, even a 1 ms difference between the slowest and second-slowest GPUs suffices for StragglAR to achieve its ideal $2\times$ speedup on a DGX H100 for a 1 GB buffer (Fig. 7(a)).

**Clarifying the end-to-end experiments [Reviewer 8vuX]**

We clarified that our end-to-end experiments do _not_ inject straggler delays and feature organic, naturally arising straggler delays.

**Heuristic vs. fundamental contribution [Reviewer ihtY]**

Reviewer ihtY claims that our approach is a “heuristic application of overlap.” We respectfully disagree. Unlike standard compute-communication overlap at the operator level, our approach enables overlap _within the communication algorithm_. Leveraging straggler delays in AllReduce is not as simple as partitioning and overlapping part of an existing bandwidth-optimal algorithm, as those algorithms rely on every GPU participating in every round to remain optimal. In fact, a heuristic-based overlap approach significantly underperforms, as we show with the “Broadcast” baseline. Instead, StragglAR completely redesigns the algorithm so that not all GPUs participate from the beginning, surpassing the bulk-synchronous lower bound when straggler delays are present and achieving competitive worst-case bounds in the absence of delay.

---

### Meta-Review · Area_Chair_jfSe · 2025-12-09

**Summary:**

Reviewers were split 2-2. The major concerns had to do with experiment scale (experiments only went up to 3B parameters) and whether the experimental conditions (including the straggler distribution) are sufficiently realistic.

**Reviewer Concerns:**

Experiment scale
- This was not addressed, as the authors did not provide new experiments on larger models. However, a 7B model could have been fine-tuned on the same 8x80GB GPUs used in the paper. Given that it is reasonable to fine-tune a 7B model, I am concerned that the lack of 7B experiments may be due to poor performance that the authors did not wish to discuss.

Experimental conditions
- The authors provided an extensive discussion on the experimental conditions; however, for systems work like this paper, it is critical to supplement such discussion with experiments and system profiling (which were not provided). I consider this issue to still be outstanding.

**Reviewer Scores:**

8vuX: their experiment scale concern was not addressed - I doubt they would have raised their score.

GAq1, HXX7: their scores were already positive. Since no new experiments were provided, I don't think their scores would have changed.

ihtY: there were concerns about this review, please see my further remarks in the appropriate section.

As a systems work, this paper really needs a 7B experiment, which is fine-tunable on the 8x80GB GPU testbed used in the paper. The omission of a 7B experiment creates a concern that it may have been withheld because the results were not consistent with the 3B model.

---

### Decision · Program_Chairs · 2026-01-26

Reject